# Impact of community-based health insurance in low- and middle-income countries: A systematic review and meta-analysis

**Paul Eze**[1]*, **Stanley Ilechukwu**[2,3], **Lucky Osaheni Lawani**[4]

1 Department of Health Policy and Administration, Penn State University, University Park, PA, United States of America, 2 Department of Global Health and Development, London School of Hygiene and Tropical Medicine, London, United Kingdom, 3 Health Projects, South Saharan Social Development Organization (SSDO), Independence Layout, Enugu, Nigeria, 4 Institute of Health Policy, Management and Evaluation, University of Toronto, Toronto, Canada

* peze@psu.edu

## Abstract

### Background

To systematically evaluate the empirical evidence on the impact of community-based health insurance (CBHI) on healthcare utilization and financial risk protection in low- and middle-income countries (LMIC).

### Methods

We searched PubMed, CINAHL, Cochrane CENTRAL, CNKI, PsycINFO, Scopus, WHO Global Index Medicus, and Web of Science including grey literature, Google Scholar®, and citation tracking for randomized controlled trials (RCTs), non-RCTs, and quasi-experimental studies that evaluated the impact of CBHI schemes on healthcare utilization and financial risk protection in LMICs. We assessed the risk of bias using Cochrane's Risk of Bias 2.0 and Risk of Bias in Non-randomized Studies of Interventions tools for RCTs and quasi/non-RCTs, respectively. We also performed a narrative synthesis of all included studies and meta-analyses of comparable studies using random-effects models. We pre-registered our study protocol on PROSPERO: CRD42022362796.

### Results

We identified 61 articles: 49 peer-reviewed publications, 10 working papers, 1 preprint, and 1 graduate dissertation covering a total of 221,568 households (1,012,542 persons) across 20 LMICs. Overall, CBHI schemes in LMICs substantially improved healthcare utilization, especially outpatient services, and improved financial risk protection in 24 out of 43 studies. Pooled estimates showed that insured households had higher odds of healthcare utilization (AOR = 1.60, 95% CI: 1.04–2.47), use of outpatient health services (AOR = 1.58, 95% CI: 1.22–2.05), and health facility delivery (AOR = 2.21, 95% CI: 1.61–3.02), but insignificant increase in inpatient hospitalization (AOR = 1.53, 95% CI: 0.74–3.14). The insured households had lower out-of-pocket health expenditure (AOR = 0.94, 95% CI: 0.92–0.97), lower

**Data Availability Statement:** All relevant data are within the paper and its Supporting Information files.

**Funding:** The author(s) received no specific funding for this work.

**Competing interests:** The authors have declared that no competing interests exist.

incidence of catastrophic health expenditure at 10% total household expenditure (AOR = 0.69, 95% CI: 0.54–0.88), and 40% non-food expenditure (AOR = 0.72, 95% CI: 0.54–0.96). The main limitations of our study are the limited data available for meta-analyses and high heterogeneity persisted in subgroup and sensitivity analyses.

## Conclusions

Our study shows that CBHI generally improves healthcare utilization but inconsistently delivers financial protection from health expenditure shocks. With pragmatic context-specific policies and operational modifications, CBHI could be a promising mechanism for achieving universal health coverage (UHC) in LMICs.

## Introduction

Starting in the 2000s, low- and middle-income countries (LMIC) implemented health system reforms aimed at improving healthcare access and outcomes to achieve universal health coverage (UHC)–a situation where all people have access to the health services they need, when and where they need them, without financial hardship [1, 2]. To this end, there has been significant interest in expanding the breadth and depth of health insurance schemes including social health insurance (SHI), national health insurance, community-based health insurance (CBHI), and private health insurance (PHI) [3, 4]. Due to the widespread interest in expanding health insurance coverage, there has also been a contemporaneous interest in evaluating the impacts of health insurance programs on the following key UHC objectives: healthcare utilization, out-of-pocket spending, and health outcomes [5, 6].

CBHI–also known as micro health insurance or mutual health insurance–are voluntary schemes characterized by community members pooling funds to offset the cost of illness and improve access to quality health services for low-income rural households largely excluded from formal health insurance schemes [7, 8]. The following institutional design features generally characterize CBHI schemes: pooling of health risks and funds occurs within a community or a group of people who share common characteristics, such as occupation or geographical location; membership premiums are often offered at a flat rate and independent of individual health risks; entitlements to benefits are linked to contributions in most cases; affiliation is voluntary; and such schemes mostly operate on a non-profit basis [9]. China's and Rwanda's remarkable strides toward UHC through the roll-out of CBHI schemes exemplify CBHI's potential for resource-limited countries seeking to achieve UHC [10].

Recent LMIC studies suggest financial barriers to healthcare, especially inpatient and specialist care and high catastrophic out-of-pocket (OOP) incidence, persists [11–14]. Previous reviews accessed the impact of CBHI schemes on healthcare utilization and/or financial protection in developing, low- and/or middle-income countries with inconsistent conclusions [15–20]. Since the publication of these reviews, several published studies that addressed these issues still inconsistently report on the impact of these schemes on UHC outcomes. Given the number of these studies, a meta-analysis would be more appropriate. We focused on CBHI's impact on healthcare utilization and financial risk protection, as achieving these critical UHC goals is pivotal to improving overall health outcomes and continued participation in the schemes [20]. The objectives of this study, therefore, are two-fold. First, we update the literature on the impact of CBHI on these key UHC objectives and discuss their implications for achieving UHC in these countries. Second, we conduct a meta-analysis of similar studies to

provide more robust estimates of the impacts of CBHI on healthcare utilization and financial risk protection in LMICs. Our main contribution is its comprehensive and rigorous evaluation of the causal evidence for CBHI's impact in this setting.

## Methods

The study protocol was prospectively published on PROSPERO: CRD42022362796; and the findings are reported according to the Preferred Reporting Items for Systematic Reviews and Meta-Analyses (PRISMA) guidelines [21].

### Eligibility criteria

We defined CBHI as the application of the principles of insurance by a defined community in a way unique to their cultural and social context, as directed by a community's choice and based on their arrangement and structures [9, 22]. Thus, we considered mutual health insurance, mutual health organizations, micro health insurance, rural health insurance, community health funds, and community health prepayment scheme as synonyms. To be included, a study must report the impact of CBHI on healthcare utilization and/or financial risk protection (**S1 Table**). However, the estimation of the impact of CBHI schemes using non-experimental data is complicated by endogeneity–heterogeneity in unobservable individual characteristics of the scheme enrollees and non-enrollees, which influences the decision to participate in the scheme and our study outcomes–pertaining to healthcare utilization and health expenditures [23, 24]. Hence, in addition to randomized control trials (RCTs) and non-RCTs, we only included studies that used statistical methods to simulate exogenous variation in the exposure to CBHI to identify and estimate causal effects [25] –**S1 Table**.

### Search and identification strategy

We searched PubMed (MEDLINE), CINAHL, Cochrane CENTRAL, ECONLIT, Embase, CNKI, PsycINFO, Scopus, Web of Science, and Global Health Library indexes (African Index Medicus, Index Medicus for the Eastern Mediterranean Region, Index Medicus for the South-East Asia Region, Literatura Latino-Americana e do Caribe em Ciências da Saúde [LILACS], and Western Pacific Region Index Medicus). We also searched ELDIS (Institute of Development Studies), IDEAS/RePEc, and 3ie impact evaluation databases. We supplemented these with a search of (1) grey literature websites–New York Academy of Medicine Grey Literature and Open Grey; (2) preprints–Gates Open Research, medRxiv, PrePubMed, Research Square, SSRN, and Wellcome Open Research; (3) websites of the World Bank, World Health Organization WHOLIS database, USAID, Inter-American Development Bank, Global Development Network, National Bureau of Economic Research, and RAND Corporation; (4) ProQuest database for dissertations & theses; and (5) Google Scholar. Finally, we tracked included studies' backward and forward citations.

We (PE and LOL) searched each database and website from its inception to 30 September 2022 using search relevant Medical Subject Headings (MeSH) terms–community-based health insurance, catastrophic health expenditure, financial risk protection, healthcare utilization, low- and middle-income countries, and developing countries from 04 to 17 October 2022 (**S1 Text**). We also used Boolean operators "AND" and "OR" to broaden the search. We sought evidence on the impact of CBHIs on healthcare utilization and financial risk protection derived from robust quantitative analysis of household or individual-level data. We considered studies published in any of the six United Nations (UN) languages–Arabic, Chinese, English, French, Russian and Spanish–and translated non-English publications using a translation service. Furthermore, we conducted a moderation exercise to ensure the eligibility criteria were

uniformly applied to article selection before independently assessing the titles and abstracts. We retrieved and assessed the full texts of eligible studies against the inclusion criteria. At every stage, we resolved discrepancies through discussion. We used Mendeley Desktop® to identify and remove duplicates.

## Data extraction

We (PE and SI) independently extracted data from the included studies using a template. We extracted the following data from each included study: authors' names, publication status, publication year, study setting, study design, data source, study (data collection) period, sampling method, sample size, statistical analysis approach, and the effect estimate of CBHI on healthcare utilization and financial risk protection. We extracted the reported effect estimate with the 95% confidence interval or standard error at 5.0% statistical significance. In cases where two or more studies used the same secondary data to estimate the impact of the same CBHI scheme, we assessed the peer-review status of the studies, prioritizing peer-reviewed studies over non-peer-reviewed studies. In addition, we extracted outcome data for all thresholds where a study described outcome measures using more than one CHE definition. We extracted CBHI effect estimates on OOP payments or CHE incidence measured using incurred medical expenditure only [26].

While community involvement in the scheme's management is common in all CBHI schemes, the degree of involvement varies from one scheme to the next. Therefore, based on the detailed description by Bennet et al. and Mebratie et al. [20, 27], we categorized CBHI schemes into *community-driven and community-managed schemes* where the community manages and administers the scheme even if the schemes was initiated by government, an NGO, or donors; *provider-based health insurance schemes* where provider, usually a hospital, plays a foremost role in mobilizing the community and community's role is more supervisory; or *government-supported community-involved schemes* which is characterized by strong government supervision and involvement. We grouped study countries into six World Bank regions (East Asia and Pacific, Europe and Central Asia, Latin America and the Caribbean [LAC], Middle East and North Africa [MENA], South Asia, and Sub-Saharan Africa [SSA]) and three income categories (low, lower-middle, and upper-middle) based on the World Bank's classification [22]. In the case of panel studies and repeated surveys that spanned multiple years, we assigned the study's last year as the year of study. We prioritized outcome measures from intention-to-treat analysis for RCTs [28], and nearest-neighbor matching for non-randomized studies employing propensity score matching [29]. We contacted the study authors to request missing estimates and/or further analysis. In addition, we resolved discrepancies through discussion.

## Risk of bias assessment

We (PE and SI) independently used Cochrane's Risk of Bias 2.0 (RoB 2.0) tool to assess the risk of bias RCTs and their respective protocols and trial registry records [30] in five domains: (1) bias arising from the randomization process, (2) bias due to deviations from intended interventions, (3) bias due to missing outcome data, (4) bias in the measurement of the outcome, and (5) bias in the selection of the reported result. If any of the five domains were associated with some concerns of risk of bias or high risk of bias, then we rated the overall risk of bias as "some concern" or "high risk", respectively. Otherwise, we rated RCTs as "low risk". Likewise, we independently assessed the risk of bias in non-RCTs and quasi-experimental studies across seven domains using the Cochrane Risk Of Bias In Non-randomized Studies of Interventions (ROBINS-I) tool [31]. We rated the overall risk of bias as "low risk", "moderate

risk", or "serious/critical risk". We resolved discrepancies through discussion. We graphically presented the risk of bias assessment for RCTs and quasi/non-RCTs using the Risk-Of-Bias VISualization (ROBVIS) tool [32].

## Data analysis

We performed narrative synthesis and meta-analysis following the Cochrane Handbook for Systematic Reviews of Interventions guidelines [33]. We used descriptive statistics to summarize the characteristics of included studies. We conducted a narrative synthesis of CBHI-impact data in included studies considering three possible effects: positive effect, statistically insignificant effect, and negative effect with relevant effect size. We performed pairwise meta-analyses using random-effects (DerSimonian-Laird) model to obtain pooled estimate of the impact of CBHI on healthcare utilization and financial protection for CBHI-insured households versus uninsured households. Multiplicity of empirical methods and outcome measures reported across included studies did not allow pooling financial risk protection outcome data in a global meta-analysis. Instead, we performed multiple meta-analyses using widely recognized measures of financial risk protection: OOP health expenditure, 10% of total expenditure, and 40% of non-food expenditures (defined also as 'consumption expenditure') [11, 26, 34–36]. For both healthcare utilization and financial protection meta-analyses; we performed "leave-one-study" sensitivity analysis to assess the impact of the different studies on the pooled estimate and sub-group analysis to assess the impact of intervention and study characteristics on pooled estimates. We assessed heterogeneity between studies using $\chi^2$ test with Cochran's Q statistic and quantified with $I^2$. Our unit of analysis was the household. We conducted statistical analyses using Stata MP 17.0 (StataCorp LLC®) and considered α (alpha) of 0.05 as cut-off for statistical significance. Map was created using QGIS version 3.28 package. Finally, we assessed the quality of evidence using the Grading of Recommendations, Assessment, Development and Evaluation (GRADE) approach for pooled estimates [37].

## Results

### Identification of studies

The study selection process is illustrated in a PRISMA flow diagram (**Fig 1**). Our literature searches identified 16,039 studies, out of which 3,431 duplicates were removed, and 12,608 studies were screened for relevance. On applying the selection criteria, 12,459 studies were excluded. Finally, 149 full texts articles were assessed and further screened using the predesigned selection criteria. Sixty-one studies met the inclusion criteria for data extraction and were included in the review [38–98], whereas 88 studies were excluded for the following reasons: the study employed an ineligible identification strategy (n = 63) [99–161], reported data from a sample already included in the review (n = 11) [162–172], case studies, reviews (n = 7) [27, 173–178], the evaluated insurance scheme is not a CBHI (n = 6) [179–184], and could not isolate the impact of CBHI scheme (n = 1) [185] –**S2 Table**.

### Characteristics of included studies

Sixty-one studies, presented in **Table 1**, were included in this review: 11 RCTs, six non-RCTs, and 44 quasi-experimental studies which compromised 1,012,542 individuals in 221,568 households across 20 LMICs (**Fig 2**). The included studies consist of 49 peer-reviewed publications, 10 working papers, one preprint, and one graduate thesis–**Table 2**. Together, the studies evaluated 63 distinct CBHI schemes: 10 government-supported community-involved, 51 community-driven and community-managed, and two provider-based CBHI schemes. The

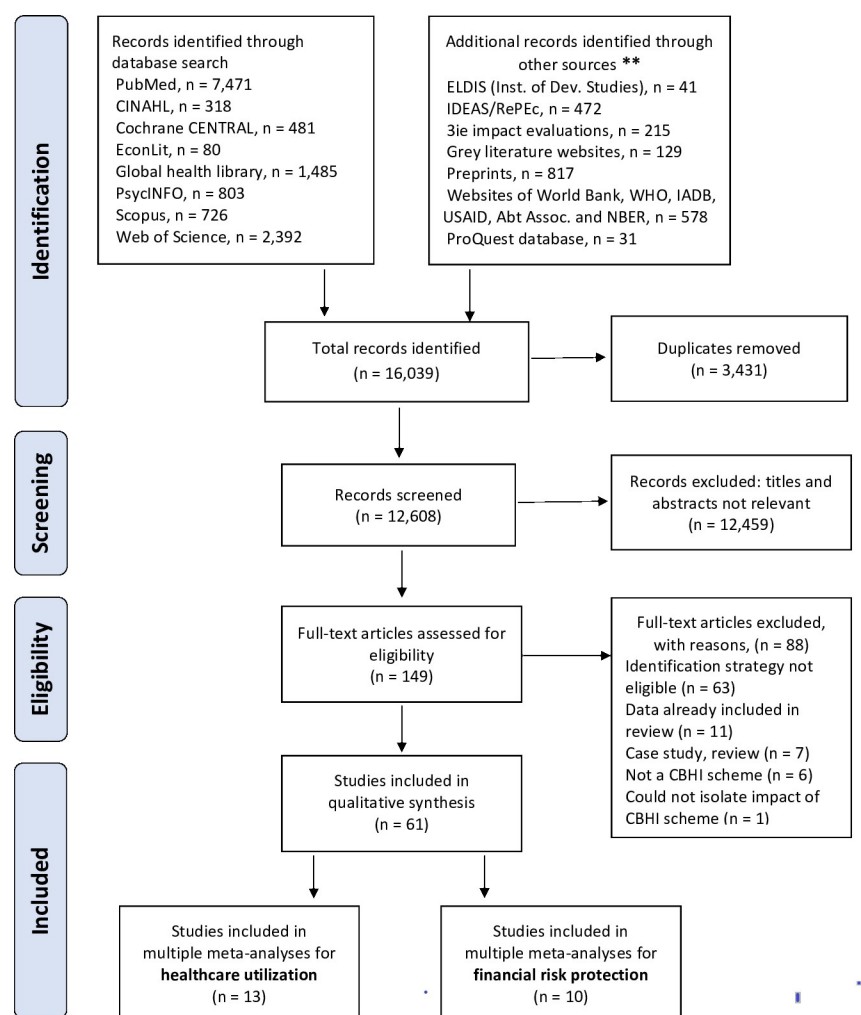

**Fig 1. PRISMA flow chart of the study identification process.** \*\* Details are provided in S2 Text.

primary studies were published between 1995 to 2022; and were undertaken in East Asia (n = 19), South Asia (n = 9), and sub-Saharan Africa (n = 33). Notably, we did not find any eligible studies in the Europe and Central Asia, LAC, and MENA regions. The median (IQR) period between the CBHI scheme's launch and study assessment was 66 (30 to 114) months and the median (IQR) coverage was 49% (18.5% to 85.5%).

Of the 11 RCTs, most (n = 10, 91%) were rated as having low overall risk of bias, and one study was rated as having some concerns [54] (**S1 Fig**). Most of the included RCTs (n = 10; 91%) were rated as having a low risk of bias arising from the randomization process, whereas the remaining RCT was rated as having some concerns for this domain. Most RCTs (n = 8; 73%) had a low risk of bias due to deviation from the intended interventions, but three had some concerns in this domain [54, 79, 82]. However, for two of these three studies, we did not consider this deviation to substantially affect the overall risk of bias in the study [79, 82]. Based on weighted risk using trials' sample size (in households), 95% of the included RCTs were rated as having a low risk of bias and approximately 5% high risk of bias (**S2 Fig**).

Of the 50 non-RCTs and quasi-experimental studies, most (n = 40; 80%) were rated as having a low overall risk of bias, eight were rated as having moderate overall risk, and two as

**Table 1. Description of community-based health insurance (CBHI) schemes in LMICs reported in included studies.**

| Study | Country | CBHI scheme (Year of launch) | CBHI model | CBHI scheme beneficiaries | Coverage rate | Health benefits covered by CBHI scheme |
|---|---|---|---|---|---|---|
| Adinma et al. 2011 [38] | Nigeria | Anambra State Government–Community Healthcare co-financing scheme (2003) | Government-supported community-involved | Residents of Igboukwu (Intervention) and Ekwulobia (Control), Aguata LGA., Anambra State, southeast Nigeria | 100% | All healthcare services with emphasis on maternal and child healthcare services amongst other health services |
| Aggarwal 2010 [39] | India | Yeshasvini health financing insurance programme (2003) | Government-supported community-involved | Rural households in Karnataka State, southwest India comprising of rural farmers and informal sector workers. | 15% | Only surgical procedures–a high-cost low-probability highly catastrophic medical event. Yeshasvini does not cover non-surgical inpatient admissions or outpatient services |
| Ahmed et al. 2018 [40] | Bangladesh | Labour Association for Social Protection-implemented community health insurance for informal low-income workers (2013) | Community-driven and community-managed | Informal workers with low income and their household members in Chandpur sub-district (comprising urban and rural areas) of Bangladesh | NR | Health benefits include GP consultations, medicines, diagnostic tests, specialist doctor's consultations, and hospitalization. Periodic satellite clinics in remote rural areas were free of charge. Benefits also includes non-health benefits such as trainings and savings opportunity |
| Alemayehu et al. 2022 [41] | Ethiopia | Ethiopia Community-Based Health Insurance (2011) | Government-supported community-involved | CBHI-enrolled households in Tigray, Amhara, Oromia, and Southern Nations Nationalities and Peoples (SNNP) regions in Ethiopia. | 49% | CBHI-members have access to healthcare services from health facilities that have agreements at the district level. Members can also access referral services when they are referred to by lower-level health facilities; otherwise, additional expenses are incurred. The benefits package does not include care abroad, private care, transportation costs, or cosmetic procedures |
| Alkenbrack & Lindelow 2015 [42] | Lao People's Dem. Rep | Community-based health insurance by the Ministry of Health with technical assistance from various donors (2001) | Government-supported community-involved | Self-employed households or those working in the informal sector in 87 villages in six districts in Vientiane Capital Province, Vientiane Province, and Champasak Province. | 10% | Outpatient and inpatient services and drugs purchased at facilities; like the benefits package covered by the formal sector schemes. |
| Ansah et al. 2009 [43] | Ghana | The Dangme West community prepayment scheme (2000) | Government-supported community-involved | Members of the Dangme West Health District in southern Ghana | NR | CBHI members receive free healthcare services including diagnosis and drugs but more limited free access to secondary health care. |
| Asfaw et al. 2022 [44] | Ethiopia | Ethiopia Community-Based Health Insurance scheme (pilot) (2011) | Government-supported community-involved | Residents of Chilga district (including Aykel, Seraba, andWohni towns), Amhara Regional State, Ethiopia. | 10% | NR |

*(Continued)*

**Table 1.** (Continued)

| Study | Country | CBHI scheme (Year of launch) | CBHI model | CBHI scheme beneficiaries | Coverage rate | Health benefits covered by CBHI scheme |
|---|---|---|---|---|---|---|
| Babatunde et al. 2016 [45] | Nigeria | Hygeia Community based Health Insurance scheme (2007) | Community-driven and community-managed | Low-income individuals in Shonga Local Government Area and its environs in Kwara State, the Lady Mechanic Initiative and Market women in Lagos State. | NR | The benefit package provides coverage for the most common medical problems that are found among the target groups and consists of Primary health care, limited Secondary care, medication and hospitalization including HIV and AIDS treatment. |
| Babiarz et al. 2010 [46] | China | New Rural Cooperative Medical Scheme (2003) | Government-supported community-involved | Rural households especially low-income farmers and informal sector workers | NR (800 million people) | Although all county programmes cover inpatient expenses, there is substantial differences between the counties. Many counties cover beyond inpatient reimbursement to include outpatient services at hospitals, township health centres, and village clinics |
| Binagwaho et al. 2012 [47] | Rwanda | Mutual health insurance (*Mutuelle de Santé*) (2005) | Government-supported community-involved | All Rwandans are eligible to enrol, but private insurance exist for the military, and formal sector workers. | 90% | Benefits include a minimum package of activities (PMA) at the primary care health centre upon a co-payment, and the complementary service package (PCA) at the district hospital with 10% co-payment. |
| Bonfrer et al. 2018 [48] | Nigeria | Kwara State Health Insurance, formerly known as Hygeia Community Health Care (2009) | Government-supported community-involved | Residents of Afon and Aboto Oja districs in Central Kwara State | 10% | KSHI covers outpatient consultations, diagnostic tests, and medication for all diseases manageable at the primary care level, as well as limited coverage of secondary care services and referral to tertiary care facilities was also covered. |
| Brals et al. 2017 [49] | Nigeria | Kwara State Health Insurance, formerly known as Hygeia Community Health Care (2009) | Government-supported community-involved | Residents of Asa Local Government Area in Kwara State. | 70.2% | Outpatient consultations, diagnostic tests, and medication for all diseases manageable at a primary care level, as well as limited coverage of secondary care services including antenatal care, vaginal and caesarean delivery, neonatal care, immunizations, radiological and more complex diagnostic tests, hospital admissions, intermediate surgery, and annual check-ups |
| Brals et al. 2019 [50] | Kenya | Community Health Plan introduced by the Africa Air Rescue Insurance, the Health Insurance Fund, and Pharm Access Foundation. (2011) | Community-driven and community-managed | Dairy farmers and their families of Tanykina Dairy Company, a cooperative of dairy farmers in rural Nandi North, Nandi County. | 100% | Benefits include all maternity services including antenatal care, delivery including caesarean section, neonatal care, and pharmacy costs for prescribed medication. Study focused on antenatal care and health facility deliveries. |

(*Continued*)

**Table 1.** (Continued)

| Study | Country | CBHI scheme (Year of launch) | CBHI model | CBHI scheme beneficiaries | Coverage rate | Health benefits covered by CBHI scheme |
|---|---|---|---|---|---|---|
| Chankova et al. 2008 [51] | Ghana Mali Senegal | Ghana Nkoranza Health Insurance Scheme (1992) Mali Four Mutual health organizations (MHO) in Bougoulaville, Wayerma, Kemeni, and Blaville (2002) Senegal 27 MHOs in Thies region, Senegal (1990) | Ghana Provider-based scheme Mali & Senegal Community-driven and community-managed | In all three countries, beneficiaries include informal sector workers and rural households. | Ghana 33% Mali 11.4% Senegal 4.8% | *Ghana*: Hospital admission and 100% refund for drugs bills during hospital admission. Outpatient care for dog and snake bite. *Mali*: Outpatient visits, hospital admission care (by one MHO), and drugs refund up to 75–80% *Senegal*: outpatient clinic visits up (varies with MHO from 50–100%), hospital admission, and essential drugs (varies with MHO from 50–100%), |
| Cheng et al. 2014 [52] | China | New Cooperative Medical Scheme (2003) | Government-supported community-involved | Rural households especially low-income farmers and informal sector workers | 96% | NCMS cover inpatient care but vary considerably in the coverage of outpatient care and reimbursement levels |
| Cheung & Padieu 2015 [53] | China | New Cooperative Medical Scheme (2003) | Government-supported community-involved | NCMS was designed to cover the whole country, but specifically for farmers and rural households | 71% | Basic healthcare but details are scanty. |
| Dercon et al. 2012 [54] | Kenya | Bima ya Jamii is a health insurance product offered by the Cooperative Insurance Company of Kenya (2010) | Community-driven and community-managed | Tea farmers living in Nyeri District, central province of Kenya, who belonged to Wananchi Savings and Credit Cooperative Society (SACCO). | 20.3% | Benefits include inpatient hospitalization cover, provided by the National Hospital Insurance Fund to all public-sector employees, as well as funeral insurance and cover for not working during hospitalization. It covers inpatient treatments and death episodes. Both insurance policies were available on the market at the time of the baseline. Despite the National Health Insurance Fund flippant decision for inpatients cover, this did not affect the availability of inpatient cover during the study. Registration was at the household level and due as a lump sum at the start of the contract. |
| Donato & Rokicki 2016 [55] | China | New Cooperative Medical Scheme (2003) | Government-supported community-involved | Rural households | 99% | Inpatient care |
| Dror et al.2016 [56] | India | 1/. Bharatiya Agro Industries Foundation (BAIF) in Pratapgarh, Uttar Pradesh (2002) 2/. Shramik Bharti in Kanpur-Dehat, Uttar Pradesh (2001) 3/. Nidan, in Vaishali, Bihar (2001) | Community-driven and community-managed | Rural and poor households in Pratapgarh, Uttar Pradesh, Kanpur-Dehat, Uttar Pradesh, and Vaishali, New Delhi | NR | Benefits packages differed and thus analysis of the data was done separately for each location. include reimbursement of healthcare bills including hospitalization costs |

(*Continued*)

**Table 1.** (Continued)

| Study | Country | CBHI scheme (Year of launch) | CBHI model | CBHI scheme beneficiaries | Coverage rate | Health benefits covered by CBHI scheme |
|-------|---------|------------------------------|------------|---------------------------|---------------|----------------------------------------|
| Fink et al. 2013 [57] | Burkina Faso | Nouna District Community-based Insurance scheme (2004) | Community-driven and community-managed | 39 villages within Nouna district in Burkina Faso. | 15.2% | Free treatment without any co-payment, ceiling, or limit on all services included in the scheme's benefit package. |
| Gnawali et al. 2009 [58] | Burkina Faso | Nouna District Community-based Insurance scheme (2004) | Community-driven and community-managed | Residents of 39 villages within Nouna district in Burkina Faso. | 15.2% | General and specialized consultation, essential and generic drugs, laboratory tests (also for antenatal care), inpatient hospital stays (up to 15 days per episode of care), X-rays, emergency surgery, and ambulance. |
| Jafree et al. 2021 [59] | Pakistan | Micro-finance provider (MFP) Health Insurance scheme (2006) | Community-driven and community-managed | Poor women for small-business development. | NR | Benefit package only covers hospital admission. |
| Jutting 2004 [60] | Senegal | 16 "Les mutuelles de sante" in the Thies region of Senegal supported by St. Jean de Dieu Hospital (1990) | Community-driven and community-managed | Rural households in Thies regions, Senegal. | 3% | CBHI member in need of surgery pay 50% of the total costs for the operation himself. The daily cost of hospitalization, including laboratory analysis and radiography, is paid by the mutual, which benefits from a 50% reduction. |
| Khan et al. 2020 [61] | Bangladesh | Labour Association for Social Protection led pilot Community-based Health Insurance (2013) | Community-driven and community-managed | Informal workers refer to the own account workers who do not have a formal job contract including rickshaw pullers, shopkeepers, restaurant workers, day labourers, factory workers and transport workers | 50% | Benefits include doctor's consultation at co-payment, Discounts for medicine and diagnostic tests, specialists' consultation, hospitalization. Non-health benefits included weekly/monthly savings opportunities and a discounted computer course |
| Kihaule 2015 [62] | Tanzania | Tanzania Community Health Fund (1999) | Government-supported community-involved | Rural households and households in the informal sector | 15% | Part payment for health expenses when sick. Details not provided. |
| Kihaule et al. 2019 [63] | Tanzania | Micro Health Insurance by the Kilimanjaro Native Cooperative Union (KNCU) and Pharm Access (2011) | Community-driven and community-managed | Coffee farmers in Moshi, Hai, and Siha in Kilimanjaro region | 27.5% | Capitation mode of payment was adopted by KNCU micro health insurance scheme to pay for health services purchased for its members which involves the payment of fixed amount of money per patient in advance to the physician by the insurer for healthcare services |

(*Continued*)

**Table 1.** (Continued)

| Study | Country | CBHI scheme (Year of launch) | CBHI model | CBHI scheme beneficiaries | Coverage rate | Health benefits covered by CBHI scheme |
|---|---|---|---|---|---|---|
| Lei & Lin 2009 [64] | China | New Cooperative Medical Scheme (2003) | Government-supported community-involved | NCMS was adopted for the households in the in rural areas where 80% of people were without health insurance of any kind. | 86% | Reimbursement of health costs differs by county. The first model, which is used by most counties (approx. 65%) involves inpatient care are reimbursed according to a formula, while outpatient services, including preventive care, are paid for through a medical savings account (MSA). The second model is used in 6.7% of counties. The inpatient reimbursement policy is the same as in the first model; but there is no MSA to cover outpatient care and preventive care usage. While the third model reimburses inpatient services as well as outpatient services for catastrophic diseases, with separate deductibles and reimbursement caps |
| Levine et al. 2016 [65] | Cambodia | Sokapheap Krousat Yeugn (SKY) Micro-Health Insurance (1998) | Community-driven and community-managed | Rural households in Takeo, Kandal, and Kampot provinces, both rural areas of Cambodia. | 2% - 12% | SKY-insured households are entitled to free services and prescribed drugs at local public health centres and at public hospitals with referral. |
| Li et al. 2019 [66] | China | New Rural Cooperative Medical Scheme (2003) | Government-supported community-involved | Rural households | 96.6% | The benefit package depends on the benefit models adopted by the county. All models involve co-payments for either outpatient visits and/or hospitalization. |
| Liu & Tsegai 2011 [67] | China | New Cooperative Medical Scheme (2003) | Government-supported community-involved | Rural and sub-urban households | 86% | Benefit package depends on the reimbursement models adopted by the county. All models involve co-payments for either outpatient visits and/or hospitalization. |
| Lu et al. 2012 [98] | Rwanda | Rwanda *Mutuelles de sante*, French for Community-based Health Insurance (1999) | Government-supported community-involved | All Rwandan households in the informal sector of the economy | 90% | CBHI-enrolled households are affiliated to designated health centres. With referrals from the health centre, members may obtain hospital services covered by *Mutuelles*. Enrollees were entitled by law to a minimum service package at the health centre and a complementary service package at the district hospital. |
| Ma 2022 [68] | China | New Rural Cooperative Medical Scheme (2003) | Government-supported community-involved | Rural households | 71% | Part-payment for medical care used. |

(*Continued*)

**Table 1.** (Continued)

| Study | Country | CBHI scheme (Year of launch) | CBHI model | CBHI scheme beneficiaries | Coverage rate | Health benefits covered by CBHI scheme |
|---|---|---|---|---|---|---|
| Mebratie et al. 2019 [69] | Ethiopia | Ethiopia Community Based Health Insurance (Pilot) scheme (2011) | Government-supported community-involved | Households in 12 districts across four regions in Ethiopia. | 48% | CBHI scheme covers all outpatient and inpatient healthcare services that are available in public facilities. Care at private providers is not covered unless a particular service or drug is unavailable at a public facility. Treatment outside the country and medical treatment with largely cosmetic value are not covered. CBHI members are exempt from co-payments if they follow the scheme's referral procedure |
| Mekonen et al. 2018 [70] | Ethiopia | Ethiopia Community-Based Health Insurance (Pilot) scheme (2011) | Government-supported community-involved | Rural households in 13 districts across the four main regions of the country (Tigray, Amhara, Oromia, and SNNP) | NR | Benefit includes both outpatient and inpatient service utilization at public facilities including food, drug, laboratory, and imaging services. Medical care in private facilities is covered if not available at public facilities. Benefits does not cover treatment abroad and plastic surgery |
| Nannini et al. 2021 [71] | Uganda | 'Doctors with Africa CUAMM' pilot Community Health Fund (CHF) (2019) | Community-driven and community-managed | Rural households in Oyam district in Northern Uganda | 17% | CHF offers her members zero-interest loan from the pooled funds to cover healthcare costs. CHF Beneficiaries have four months to repay loan at zero interest. |
| Noterman et al. 1995 [72] | Congo DRC (Formerly Zaire) | Masisi Health District Prepayment scheme (1987) | Provider-based scheme | Rural households in Masisi health district, situated in eastern Congo DR (formerly Zaire) in the mountainous Kivu region | 6.7% (1987) 26.8% (1988) | All (direct) hospital admission costs to be covered |
| Nshakira-Rukundo et al. 2021 [73] | Uganda | The Kisiizi Community Based Health Insurance scheme (1996) | Community-driven and community-managed | Rural households in five districts in south-western Uganda. | 38% | The Kisiizi scheme covers outpatient and inpatient services, surgeries, and emergency services. Investigative and imagining procedures such as X-rays, ultra-sounds, and laboratory investigations are also covered up to the full cost of the treatment. Elective surgical conditions are covered up to 50% of the cost. The insurance does not cover dental, optical procedures and self-inflicted injuries such as those arising from alcohol consumption and substance abuse |

**Table 1.** (Continued)

| Study | Country | CBHI scheme (Year of launch) | CBHI model | CBHI scheme beneficiaries | Coverage rate | Health benefits covered by CBHI scheme |
|---|---|---|---|---|---|---|
| Papoula 2012 [74] | Rwanda | Rwanda *Mutuelle de Sante* or Mutual Health Insurance Scheme (MHI) | Community-driven and community-managed | All Rwandan households in the district are eligible to enrol | 73% | Health care package offering family planning, antenatal care, consultations, normal and complicated deliveries, basic laboratory tests, generic drugs, hospital treatment for malaria and some tertiary care. |
| Parmar et al. 2012 [75] | Burkina Faso | *Assurance Maladie à Base Communautaire*–a community-based Health Insurance. (2004) | Community-driven and community-managed | Rural households in Nouna Health District (NHD) with approximately 70,000 persons. | 5% - 9% | The benefit package included a wide range of medical services available in the public health facilities in the NHD. There were no co-payments, deductibles, or ceiling on the benefit |
| Pham & Pham 2012 [76] | Vietnam | Vietnam Health Care for the Poor Program (2003) | Government-supported community-involved | Poor households, all households regardless of their own assessed income living in poor communes, and ethnic minorities living in the provinces of the central highland area and other six provinces in the North. | 60% | The CBHI scheme insurance covers the costs of both inpatient and outpatient care, and drugs used in inpatient treatment, but not non-prescription drugs. Other than antenatal care, the benefit package, however, provides limited coverage for preventive interventions. |
| Philibert et al. 2017 [77] | Mauritania | Mauritania Obstetric Risk Insurance (2002) | Government-supported community-involved | Pregnant mothers | NR | Benefits include antenatal visits, normal vaginal delivery, basic neonatal care, first-line treatment of obstetric complications, monitoring of children under 5 years and family planning. Comprehensive emergency obstetric care, including transfusions and caesarean sections at regional/national hospitals |
| Qin et al. 2021 [78] | China | New Rural Cooperative Medical Scheme (2003) | Government-supported community-involved | Rural households | NR | Part-payment for medical care |
| Ranson et al. 2007 [79] | India | Self Employed Women's Association (SEWA) Community based health insurance scheme (1992) | Community-driven and community-managed | SEWA members. SEWA is a trade union of more than half a million poor women working in the informal sector and based in the Indian State of Gujarat. | 97% | SEWA provides inpatient care coverage including hospital and provider charges, medicines, transportation, and other expenditure incurred to a maximum of 2,000 rupees a year. SEWA does not cover outpatient visits or childbirth. |
| Rao et al. 2009 [80] | Afghanistan | Afghanistan Community Health Fund (2005) | Community-driven and community-managed | Households in Parwan and Wardak provinces in central Afghanistan, Saripul province in the north, Nimroz in the south-west, and Hilmand in the south. | 6% | CHF membership covered all services offered at the designated health facility in addition to inpatient care at the nearest district hospital. |

(*Continued*)

**Table 1.** (Continued)

| Study | Country | CBHI scheme (Year of launch) | CBHI model | CBHI scheme beneficiaries | Coverage rate | Health benefits covered by CBHI scheme |
|---|---|---|---|---|---|---|
| Ravit et al. 2020 [81] | Mauritania | Mauritania Obstetric Risk Insurance (2003) | Government-supported community-involved | Pregnant mothers | 25% | Benefit includes four antenatal visits; treatments during pregnancy, blood test in each antenatal visit; one ultrasound scan during the first trimester treatment for any pathologies related to pregnancy and delivery; skilled birth delivery; treatment for complications during pregnancy & delivery, including Caesarean section; ambulance transfer to a higher-level facility; hospital care if transferred; and one postnatal care visit |
| Raza et al. 2016 [82] | India | Community-based Health Insurance schemed introduced by Delhi-based Micro Insurance Academy (2010) | Community-driven and community-managed | Households connected to self-help groups (SHGs). SHGs consist of 10–20 women living in the same village who come together and agree to save a specific amount each period in Kanpur Dehat and Pratapgarh districts in Uttar Pradesh and in Vaishali district in Bihar. | 24% | Reimbursement for outpatient, hospitalisation, and maternity care including laboratory fees, imaging, and transport. Benefit package also includes reimbursement for wage loss during hospitalization. |
| Robyn et al. 2012 (A) [83] | Burkina Faso | The *Assurance Maladie à Base Communautaire*–Community-Based Health Insurance (2004) | Community-driven and community-managed | Individuals and families from covered health areas within the Nouna demographic surveillance system. Nouna health district is a remote and rural health district situated in north-west Burkina Faso | NR | Benefits include outpatient services offered at primary-care facilities and up to 15 days inpatient care at the district hospital including essential medicines. There is no co-payment, ceiling, or limit on the number of services rendered. |
| Robyn et al. 2012 (B) [84] | Burkina Faso | The *Assurance Maladie à Base Communautaire*–Community-Based Health Insurance (2004) | Community-driven and community-managed | Interested households in Nouna Health District who paid the enrolment fee and annual premium. | 50% | Benefits include outpatient services offered at primary care facilities and up to 15 days of inpatient care at the district hospital are covered, as well as all essential medicines offered at public facilities. There is no co-payment, ceiling, or limit on number of services rendered. |

(*Continued*)

**Table 1.** (Continued)

| Study | Country | CBHI scheme (Year of launch) | CBHI model | CBHI scheme beneficiaries | Coverage rate | Health benefits covered by CBHI scheme |
|---|---|---|---|---|---|---|
| Sheth 2021 [85] | India | Chaitanya's Dipthi Arogya Nidhi (DAN) mutual health insurance scheme (Chaitanya is a non-profit Micro Finance Institution in Junnar sub-district of rural Maharashtra) (2011) | Community-driven and community-managed | Offered to interested self-help groups for women in sub-district of rural Maharashtra | 80% | Benefits include discounted prices (ranging from 5% to 20%) negotiated at private network medical facilities, (hospitals, labs, pharmacies). For inpatient hospitalization, members receive 60% reimbursement of their medical fees at network private hospitals, and full reimbursement at public medical facilities, up to a limit of Rs. 15,000 (USD 300). Benefits also includes a 24–7 medical helpline, health camps, and monthly visits by a doctor to villages to offer referrals and basic medicines |
| Shimeles 2010 [86] | Rwanda | Rwandan Community Based Health Insurance, locally referred to as Mutuelle de Santé (2004) | Government-supported community-involved | Poor households and households in the informal sectors | 85% | NR |
| Simieneh et al. 2021 [87] | Ethiopia | Ethiopia Community-Based Health Insurance scheme (2011) | Government-supported community-involved | All Ethiopian households especially rural poor households | NR | NR |
| Sun et al. 2009 [88] | China | New Cooperative Medical Scheme pilot scheme in Linyi County (2003) | Government-supported community-involved | Farmers and rural households in rural areas | 94.6% | Benefits cover hospital outpatient and inpatient services. Outpatient reimbursements averaged 20% of total expenses. Inpatients received discounts of 20–80% of total expenses; the higher the expenses, the higher the benefit up to a ceiling of 10 000 yuan per person per year |
| Tilahun et al. 2018 [89] | Ethiopia | Ethiopia Community-Based Health Insurance scheme (2011) | Government-supported community-involved | Rural poor households and households in the informal sector | 83% | Benefit package covers all outpatient and inpatient services at all levels of the health facilities except for dentures, eyeglasses, and cosmetic healthcare service |
| Wagstaff 2007 [96] | Vietnam | Vietnam Health Care Fund for the Poor (2002) | Government-supported community-involved | Poor households, disadvantaged households, and ethnic minorities living in province of Thai Nguyen and six mountainous provinces designated as facing special difficulties | 15% | The benefit package includes services delivered by public hospitals and commune health centres. The scheme does not cover non-prescription drugs bought from drug vendors and pharmacies. The package also excludes services delivered by other private providers. |

*(Continued)*

**Table 1.** (Continued)

| Study | Country | CBHI scheme (Year of launch) | CBHI model | CBHI scheme beneficiaries | Coverage rate | Health benefits covered by CBHI scheme |
|---|---|---|---|---|---|---|
| Wagstaff et al. 2009 [97] | China | New Cooperative Medical Scheme (2003) | Government-supported community-involved | Rural Chinese households | Approx 80% | Considerable heterogeneity in the benefit package between counties and coverage models. However, all the counties cover inpatient care with different co-payments rate. However, not all counties cover outpatient care or covered partially. |
| Woldemichael & Shimeles 2015 [90] | Rwanda | Rwandan Community Based Health Insurance, locally referred to as *Mutuelle de Santé* (2004) | Government-supported community-involved | Poor households and households engaged in small scale agriculture and in the informal sectors | 86% | Benefit packages include comprehensive preventive and curative healthcare services provided at local health centres and referral hospitals. These services, provided at all levels of healthcare provision in Rwanda (health centres, district hospitals, and referral hospitals), are categorized into two packages: Minimum Package of Activities (MPA) and Complementary Package of Activities (CPA). The MPA covers services provided at health centres which include promotional, preventive, and curative health services. The CPA, on the other hand, covers services provided at district hospitals. Healthcare services covered by CBHI include vaccination, medical consultation, surgery, dental care and surgery, radiology and scanning, laboratory tests, physiotherapy, hospitalization, accepted list of essential drugs available at health centres and hospitals, prenatal, perinatal, and postnatal care, ambulatory service, prosthesis and orthose |
| Woldemichael et al. 2019 [91] | Rwanda | Rwandan Community Based Health Insurance, locally referred to as *Mutuelle de Santé* (2004) | Government-supported community-involved | Households engaged in small scale agriculture and in the informal sectors | 86% | As described in Woldemichael & Shimeles 2015 above |
| Yang 2015 [92] | China | New Cooperative Medical Scheme (2003) | Government-supported community-involved | Rural Chinese households | 97.5% | NCMS benefits includes access to a range of health facilities, from village clinics to municipal hospitals, although the reimbursement rates for health services received differ from one facility to another |

(*Continued*)

**Table 1.** (Continued)

| Study | Country | CBHI scheme (Year of launch) | CBHI model | CBHI scheme beneficiaries | Coverage rate | Health benefits covered by CBHI scheme |
|---|---|---|---|---|---|---|
| Yilma et al. 2015 [93] | Ethiopia | Ethiopia Community-Based Health Insurance scheme (2011) | Government-supported community-involved | Rural poor households and households in the informal sector | 48% | Benefit package includes both outpatient and inpatient service utilization at public facilities. Enrolled households may not seek care in private facilities unless a particular service or drug is unavailable at a public facility. The scheme excludes treatment abroad and treatments with large cosmetic value such as artificial teeth and plastic surgery. |
| Yip & Hsiao 2010 [94] | China | China's Rural Mutual Health Care (RMHC) (2003) | Community-driven and community-managed | Rural Chinese households | 90% | RMHC covered all primary care, hospital services, and drugs at all levels of facilities with no deductibles with various rates of co-insurance. |
| Yip et al. 2008 [95] | China | Rural Mutual Health Care scheme (2003) New Cooperative Medical Scheme model in the Western and Central Regions (2003) | Community-driven and community-managed Government-supported community-involved | Rural Chinese households | 50% | The RMHC reimbursed 45% of outpatient visits expenses and 40% of hospitalization costs The NCMS in the Western and Central Regions only reimbursed 50% of hospitalization costs |

**CBHI**: Community-based Health Insurance, **ORS**: Oral Rehydration Salt

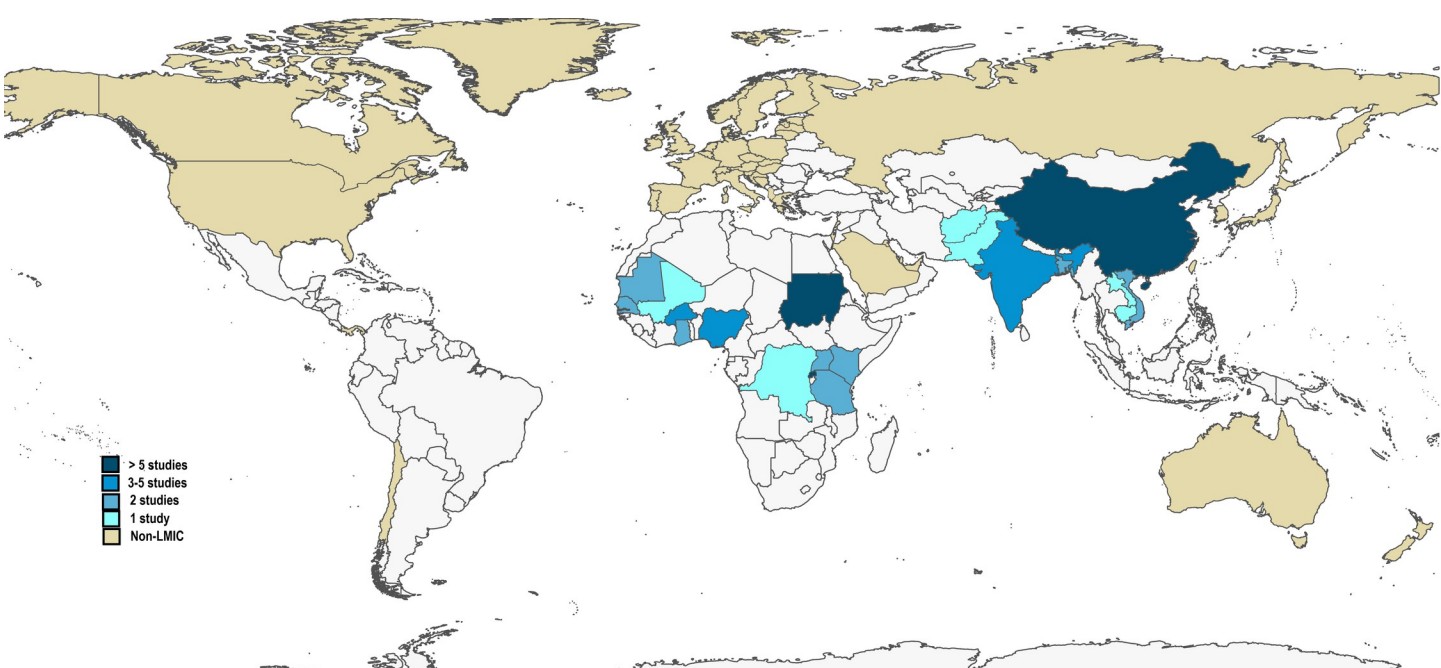

**Fig 2. Choropleth map of countries represented in the systematic review according to the number of studies in which each country is represented.** The base layer map is obtained in QGIS version 3.28 (Firenze) software, which imports the world map from Natural Earth, which is in the public domain and available from https://www.naturalearthdata.com; with terms of use available in http://www.naturalearthdata.com/about/terms-of-use/.

**Table 2. Summary of the reported impact of CBHI schemes on healthcare utilization and financial risk protection from included studies.** Adjusted effect estimates (with 95% confidence interval, where applicable) in bold.

| Study Publication status | Study design | Data sources (Data year) | Sample size | Identification strategy | Key findings on the impact on healthcare utilization | Key findings on the impact on financial protection |
|---|---|---|---|---|---|---|
| Adinma et al. 2011 [38] *Published* | Pre-post intervention controlled | Repeat cross-sectional data (2004–2005) | 240 | Regression | CBHI significantly increased use of healthcare services. Health facility delivery: **AOR = 2.486 (2.031–2.979)** | NR |
| Aggarwal 2010 [39] *Published* | Cross-sectional | Household survey (2007) | 4,109 (21,630 persons) | Propensity score matching (PSM) and Regression | Those with health insurance used health facilities in greater numbers and with greater frequency; increase from outpatient service usage, including outpatient surgery; Insignificant increase in utilization of in-patient care (hospitalization). | Overall health expenditures are 19–20% higher for Yeshasvini enrollees compared with uninsured cooperatives. However, enrollees enjoyed substantial financial protection by reducing the need to borrow money or sell assets to meet the medical expenses. |
| Ahmed et al. 2018 [40] *Published* | Cross-sectional | Household survey (2014) | 1,292 | PSM and Regression | Use of medically trained healthcare providers was higher amongst the CBHI enrollees versus non-insured. **AOR = 2.111 (1.458–3.079).** CBHI scheme increases the utilization of medically trained healthcare providers among informal workers. | NR |
| Alemayehu et al. 2022 [41] *Preprint* | Case-comparison | Household survey (2020) | 4,238 | PSM and Regression | CBHI membership is significantly associated with a higher number of per capita OPD visits but inpatient admission was lower among CBHI enrollees than non-CBHI enrollees. | Direct out-of-pocket medical expenses were lower among CBHI member households compared to non-member households both for outpatient and inpatient services CBHI enrolment was significantly associated with lower likelihood of catastrophic health spending **10% THE AOR = 0.504 (0.375–0.677) 40% NFE AOR = 0.356 (0.204–0.621)** Impoverishment due to OOP health payments among CBHI member households was lower compared to households from non-CBHI implementing woredas but higher compared to non-CBHI member households from CBHI implementing woredas |

*(Continued)*

**Table 2.** (Continued)

| Study Publication status | Study design | Data sources (Data year) | Sample size | Identification strategy | Key findings on the impact on healthcare utilization | Key findings on the impact on financial protection |
|---|---|---|---|---|---|---|
| Alkenbrack & Lindelow 2015 [42] *Published* | Mixed methods: Cross-sectional case-comparison with qualitative component | Household survey (2009) | 3,000 (14,804 persons) | PSM and Regression | CBHI significantly increased utilization of both inpatient and outpatient services. CBHI members did not make more inpatient visits than non-members, but they did make more outpatient visits. CBHI members were more likely to use public health facilities and less likely to use private clinics. | CBHI enrollees were less likely to incur catastrophic health expenditures than non-enrollees. **10% THE threshold AOR = 0.645 (0.477–0.874)** Although the CBHI scheme provides some protection against catastrophic expenditures, qualitative data suggests some CBHI households still incur catastrophic expenditures, especially those with inpatient visits for services not covered by the scheme. |
| Ansah et al. 2009 [43] *Published* | Randomized controlled trial (RCT) | Household survey (2004 and 2005) | 2,194 | Regression | CBHI has a significant impact on health care utilisation. Children were taken to primary care facilities significantly more frequently in the intervention arm vs. than in the control arm. **RR = 1.12 (1.04–1.20; p = 0.001).** | NR |
| Asfaw et al. 2022 [44] *Published* | Cross-sectional | Household survey (2022) | 531 | PSM and Regression | CBHI-enrolled households used outpatient and inpatient services (visits) 2.6 times compared to the non-enrolled households **ATT = 2.62 (SE = 0.152)** | CBHI-enrolment significantly reduced household per-capita health expenditure by 17 percentage points compared to non-enrolled. **ATT = -0.17 (SE = 0.013)** |
| Babatunde et al. 2016 [45] *Published* | Cross-sectional | Household survey (2014) | 175 | PSM and Regression | NR | CBHI enrolment significantly reduced out-of-pocket payments and increased per capita income by N1,163.44 |
| Babiarz et al. 2010 [46] *Published* | Cross-sectional | Household survey (2004 and 2008) | 2,800 | Difference-in-differences | No single policy feature of NCMS was associated with statistically significant changes in the probability of medical care use when sick or with the choice of where to use services | NCMS reduced individual patients' health expenditures (particularly outlier out of pocket spending) and the need to borrow or sell assets to pay for medical care |
| Binagwaho et al. 2012 [47] *Working Paper* | Cross-sectional | Rwandese Demographic and Health Surveys (2005 and 2010) | 8,384 | Instrumental variable (IV) and Regression | CHBI-enrolled children were more likely to access modern healthcare compared with children in uninsured households **AOR = 1.175 (1.095–1.261)** | NR |

(*Continued*)

**Table 2.** (Continued)

| Study Publication status | Study design | Data sources (Data year) | Sample size | Identification strategy | Key findings on the impact on healthcare utilization | Key findings on the impact on financial protection |
|---|---|---|---|---|---|---|
| Bonfrer et al. 2018 [48] *Published* | Controlled interrupted time-series | Household panel data (2009 and 2011) | 3,509 | PSM and Difference-in-differences | CBHI-enrolment led to a statistically significant increase (by 36%, p < 0.000) in formal healthcare utilization compared to the uninsured | CBHI-enrolment reduced out-of-pocket expenditure (63%, p < 0.000) significant reduction in per capita health expenditure for the insured vs. uninsured. |
| Brals et al. 2017 [49] *Published* | Controlled interrupted time-series | Household panel data (Baseline in 2009; follow-up surveys in 2011 & 2013) | 1,500 | PSM and Difference-in-differences | CBHI enrolment led to the increase in hospital deliveries by 29.3 percentage points (95% CI: 16.1 to 42.6; P < 0.001) greater than the change in the control area, corresponding to a relative increase in hospital deliveries of 62% | NR |
| Brals et al. 2019 [50] *Published* | Quasi-experimental controlled before-after | Household panel data (Baseline in 2011; follow-up surveys in 2014) | 295 | Difference-in-differences and Regression | Antenatal care utilization significantly increased after TCHP was introduced (14.4 percentage points; 95% CI: 4.5–24.3; p = 0.004) but no significant effect was observed on health facility deliveries (8.8 percentage points; 95% CI: -14.1 to +31.7; p = 0.450). | NR |
| Chankova et al. 2008 [51] *Published* | Cross-sectional | Household survey (2004) | Ghana 1,806 Mali 2,659 Senegal 1,080 | Instrumental variable (IV) and Regression | CBHI enrolment had mixed impact on healthcare utilization in the three countries. Sought modern curative care **Ghana AOR = 1.81 (1.101–2.975) Mali AOR = 0.90 (0.512–1.581) Senegal AOR = 1.48 (1.068–2.051)** Hospitalization **Ghana AOR = 1.09 (0.786–1.512) Senegal AOR = 2.28 (1.338–3.884)** | CBHI enrolment had mixed impact on households' exposure to catastrophic health spending compared to the uninsured. **Mali AOR = 1.094 (0.891–1.343) Senegal AOR = 0.619 (0.207–1.853)** |

*(Continued)*

**Table 2.** (Continued)

| Study Publication status | Study design | Data sources (Data year) | Sample size | Identification strategy | Key findings on the impact on healthcare utilization | Key findings on the impact on financial protection |
|---|---|---|---|---|---|---|
| Cheng et al. 2015 [52] *Published* | Cross-sectional | Chinese Longitudinal Healthy Longevity Survey (2005 and 2008) | 3,299 | PSM and Difference-in-differences | NCMS significantly increased the probability of getting adequate care when sick by 5.5 percentage points (pp) and reduced the risk of failing to get necessary care due to costs by 3 pp. | There is no evidence to suggest NCMS reduced enrollee's total medical expenditure and out-of-pocket expenditure. Instead, the estimates are positive but insignificant. |
| Cheung & Padieu 2015 [53] *Published* | Cross-sectional | China Health and Nutrition Survey (2006) | 1,312 | PSM, Instrumental variable, and Regression | NR | Higher middle income participants deplete their savings significantly compared to non-participant households. Higher middle-income participants save less than non-participants. There was no financial protective impact of the health care scheme on the poorest, and richest are unaffected. |
| Dercon et al. 2012 [54] *Working paper* | Cluster RCT | Household panel survey (2009–2012) | 1,500 | Regression | The MHI does not have any statistically significant impact on increasing the probability of going to the hospital when sick. | MHI has positive but insignificant impact on net health expenditures, informal borrowing for medical costs, food consumption, non-food consumption, and overall consumption |
| Donato & Rokicki 2016 [55] *Working paper* | Longitudinal cross-sectional study | China Health and Nutritional Survey (2000–2009) | 4,563 | Instrumental variable | Each additional year of NCMS coverage was associated with an increase in the probability of use of preventive care by 0.6 pp (95% CI = 0.1 to 1.0), with was some suggestive evidence of improvements for relatively poor families. | NR |
| Dror et al.2016 [56] *Published* | Cluster RCT | Household surveys and Management Information System. (2010–2013) | 3,307 (18,322 persons) | PSM and Difference-in-differences analysis | The positive attitude of seeking medical help went together with the length of period of insurance. | Insured households reported significantly less borrowing for non-hospitalization events than uninsured households over time. The probability insured families to benefit from quintile downgrade decreased |

(*Continued*)

**Table 2.** (Continued)

| Study Publication status | Study design | Data sources (Data year) | Sample size | Identification strategy | Key findings on the impact on healthcare utilization | Key findings on the impact on financial protection |
|---|---|---|---|---|---|---|
| Fink et al. 2013 [57] *Published* | Stepped-wedge cluster RCT | Nouna Health and Demographic Surveillance Site (NHDSS) household panel survey in each year (2004–2006) | 1,530 | Regression | NR | CBHI had limited effects on average OOP expenditures, but substantially reduced the likelihood of catastrophic health expenditure **5% THE AOR = 0.728 (0.385–1.071) 10% THE AOR = 0.830 (0.546–1.114) 15% THE AOR = 0.859 (0.616–1.102) 25% THE AOR = 0.876 (0.659–1.094) 50% THE AOR = 0.958 (0.811–1.105)** |
| Gnawali et al. 2009 [58] *Published* | Stepped-wedge cluster RCT | Nouna Health and Demographic Surveillance Site (NHDSS) household panel survey (2004–2006) | 1,309 | Regression | CBHI-insured individuals were more likely to seek modern healthcare when sick at the health facility than the uninsured (46.5% vs 18.2%). **AOR = 3.985 (2.474–6.132)**, but had insignificant effect on inpatient hospitalization | NR |
| Jafree et al. 2011 [59] *Published* | Cross-sectional | Survey data (2018) | 442 | PSM and Regression | Health micro-insurance did not have significant positive effect on ability to visit general practitioner (**Coeff. = 0.0038, p = 0.968**), or pay for prescribed medicines (**Coeff. = 0.0141, p = 0.818**). | NR |
| Jutting 2004 [60] *Published* | Cross-sectional | Household survey (2000) | 346 | PSM and regression | CBHI members are 2.0 percentage points more likely to use hospital care than non-members (marginal effect). | CBHI members have out-of-pocket payment for hospital care decreased by 45.2 percent in comparison to non-members |
| Khan et al. 2020 [61] *Published* | Case comparison | Household survey (2014) | 1,292 | PSM and Regression | NR | CBHI-enrolled members had significantly less likelihood of incurring out-of-pockets payments, by 6.4 percentage points, than the uninsured. **AOR = 0.938 (0.913–0.964)** |
| Kihaule 2015 [62] *Published* | Cross-sectional | Tanzania Demographic and Health Survey (2011) | 10,300 | PSM | NR | CBHI scheme protected against catastrophic health spending, in the episodes of illness to members of insurance schemes compared to non-members. **(ATE = -0.072, p = 0.035)** |
| Kihaule et al. 2019 [63] *Published* | Cross-sectional | Household survey (2016) | 1,080 | PSM | About 1.6 percentages points probability of higher health services utilization among CBHI members compared to non-members. | CBHI-enrolment did not offer any significant protection from catastrophic health spending compared to non-members (**ATT difference = 0.644, p = 0.229**). |

*(Continued)*

**Table 2.** (Continued)

| Study Publication status | Study design | Data sources (Data year) | Sample size | Identification strategy | Key findings on the impact on healthcare utilization | Key findings on the impact on financial protection |
|---|---|---|---|---|---|---|
| Lei & Lin 2009 [64] *Published* | Cross-sectional | China Health and Nutrition Survey (CHNS) data (2016) | 3,952 | Instrumental variable, fixed effects, PSM, Difference-in-differences, and Regression | NCMS significantly increases the utilization of preventive care, particularly general physical examinations **(AOR = 1.015 (1.005–1.025)**, but it does not significantly increase utilization of formal medical services **(AOR = 1.008 (0.803–1.265).** | NCMS does not decreases out-of-pocket expenditure for CBHI-insured households compared to uninsured households **AOR = 1.134 (0.592–2.174)** |
| Levine et al. 2016 [65] *Published* | RCT | Household survey (2008) | 5,008 | Instrumental variable and Regression | CBHI-insured households were 15.8 percentage points more likely to use a health center for first treatment (p < 0.001) and 10.7 and 8.0 percentage points less likely (p < 0.05 and p < 0.05, respectively) to visit a private doctor or drug seller for first treatment compared to the control group. | Compared to the uninsured, the insured were significantly less likely to pay for care using a loan (with or without interest), less likely to have increased debt, and less likely to pay for care by selling assets. The insured also had lower out-of-pocket expenditures and were less likely to have large expenditure for care |
| Li et al. 2019 [66] *Published* | Cross-sectional | China Health and Retirement Longitudinal Study (2011) | 12,561 | Instrumental variable and Regression | NCMS increased the probability of outpatient care utilization (p<0.05 for all models) particularly preventive care services. However, NCMS has no impact on inpatient care utilization (p>0.05). | NCMS had no significant impact on out-of-pocket expenditures for both outpatient and inpatient care (p>0.05 for all models). |
| Liu & Tsegai 2011 [67] *Working paper* | Case-comparison | China Health and Nutrition Survey (CHNS) data (2006) | 2,058 | PSM and bounding approach | NCMS has a statistically significant positive impact on improving outpatient utilization for both eastern and western regions, but not in the central regions. | NCMS has no statistically significant impact on reducing medical burden. NCMS rather increased the incidence of catastrophic spending for western region. |
| Lu et al. 2012 [98] *Published* | Cross-sectional | Rwanda Integrated Living Conditions Survey (2000 and 2006) Demographic Health Survey (2005 and 2008) | 2000 6,408 2006 6,280 | PSM, Instrumental variable, and Regression | CBHI enrolment was associated with higher utilization of medical care (N = 5,435) **AOR = 2.124 (1.614–2.794)** Child healthcare utilization (N = 4,421) **AOR = 2.002 (1.480–2.707)** Skilled birth attendance (N = 1,766) **AOR = 1.779 (1.188–2.664)** | CBHI enrolment was associated with lower incidence of household CHE at the 40% NFE threshold. (N = 5,430). **AOR = 0.682 (0.523–0.798)** |

*(Continued)*

**Table 2.** (Continued)

| Study Publication status | Study design | Data sources (Data year) | Sample size | Identification strategy | Key findings on the impact on healthcare utilization | Key findings on the impact on financial protection |
|---|---|---|---|---|---|---|
| Ma 2022 [68] *Published* | Cross-sectional | China Health and Nutrition Survey (2000 to 2011) | 7,200 | Difference-in-differences, and Regression | NRCMS did not affect ill persons' probabilities of visiting hospitals (either outpatient or inpatient) in the short term, after the implementation of the NRCMS (2004 or 2006). | NR |
| Mebratie et al. 2019 [69] *Published* | Cross-sectional | Household panel survey (2011–2013) | 1,632 | PSM and Regression | CBHI-enrolled households using outpatient care from health centres increases by 10 pp (from 20% in 2011 to 30% in 2013) for the insured while there is a slight decline for the control group. **AOR = 1.053 (1.014–1.093** CBHI-enrolment also led to a 45–64% increase in the frequency of healthcare use, relative to the baseline. | While the CBHI scheme reduced the cost of medical care, it is not significantly associated with a decrease in catastrophic out-of-pocket expenditure incidence (10% total household expenditure) in the insured households compared with uninsured households (p > 0.10) |
| Mekonen et al. 2018 [70] *Published* | Cross-sectional | Household survey (2016) | 454 | PSM and Regression | NR | CBHI-enrolled households had lower odds of incurring catastrophic health expenditure than uninsured households **15% NFE AOR = 0.190 (0.110–0.340)** |
| Nannini et al. 2021 [71] *Published* | Mixed methods: Cross-sectional surveys and qualitative components | Household survey (2019) | 226 | Instrumental variable regression | NR | Community-health fund led to a significant decrease in the incidence of health expenditures, share of health expenses, and of the occurrence of financial hardship, compared to the uninsured households. Among enrolled members, the poorest receive a greater benefit from the intervention compared with the well off. |

*(Continued)*

**Table 2.** (Continued)

| Study Publication status | Study design | Data sources (Data year) | Sample size | Identification strategy | Key findings on the impact on healthcare utilization | Key findings on the impact on financial protection |
|---|---|---|---|---|---|---|
| Noterman et al. 1995 [72] *Published* | Repeated interrupted time series (with control group) | Health facility registry data (1987–1990) | 1,847 | Bivariate analysis | The hospital admission rates for prepayment members were significantly higher than for non-members (p = 0.025). Travel distance had a lower impact on hospital admission for enrolled than non-enrolled. The hospital admission rate for admission rate for deliveries was almost 7 times higher in the prepayment members than non-members (p = 0.005). Almost all enrolled pregnant women delivered in the hospital. | NR |
| Nshakira-Rukundo et al. 2021 [73] *Published* | Cross-sectional | Household surveys (2015) | 464 | PSM and regression | CBHI enrolment significantly increased the probability of using long-lasting insecticide-treated nets by 25.5% and deworming by 17.5%. | NR |
| Papoula 2012 [74] *Doctoral Thesis* | Cross-sectional | Household surveys (2010) | 12,540 | PSM and regression | Across all income quintiles MHI-enrolled households had more a higher probability of visiting health care centres when sick more than 50% for both outpatient and inpatient visits. | MHI-enrolled individuals face less out-of-pocket expenses than their uninsured persons in all income quintiles apart from those belonging in the poorer quintile but the difference is insignificant. |
| Parmar et al. 2012 [75] *Published* | Cluster RCT | Household survey (2004–2006) | 890 | Instrumental variable model and Regression | NR | CBHI has a positive effect on per capital household assets, compared to uninsured. **AOR = 1.246 (0.983–1.580)** |
| Pham & Pham 2012 [76] *Working paper* | Cross-sectional | Vietnam Household Living Standard Survey VHLSS (2004, 2006, and 2008) | 8,304 | Instrumental variable and Regression | CBHI was associated with a substantial increase in the intensity of seeking health care from a public facility. | CBHI was associated with a substantial reduction in the overall OOP health spending, but insignificant impact on catastrophic health expenses **40% NFE threshold AOR = 0.899 (0.645–1.251)** |

(*Continued*)

**Table 2.** (Continued)

| Study Publication status | Study design | Data sources (Data year) | Sample size | Identification strategy | Key findings on the impact on healthcare utilization | Key findings on the impact on financial protection |
|---|---|---|---|---|---|---|
| Philibert et al. 2017 [77] *Published* | Quasi-experimental adjusted before-and-after study | Demographic and Health Survey {DHS} (2001) National Survey on Infant Mortality and Malaria {NSIMM} (2003), and Multiple Indicator Cluster Survey (MICS) (2007, 2011) | DHS 2001 7,728 NSIMM 2003 5,211 MICS 2007 12,549 MICS 2011 12,754 | Regression | ORI had significant effect on antenatal visit: **AOR = 1.53 (1.12–2.10)** but had no significant positive effect on postnatal care attendance: **AOR = 1.00 (0.73–1.36)** In contrast, caesarean delivery and modern contraceptives use significantly increased more rapidly in districts with no ORI. Caesarean delivery: **AOR = 0.42 (0.22–0.78)** and for modern contraceptive use: **AOR = 0.42 (0.26–0.68)** | NR |
| Qin et al. 2021 [78] *Published* | Cross-sectional | China Family Panel Studies (2016) | 5,877 | Instrumental variable | NR | NRCMS has significantly reduced the likely risk of falling into poverty due to hospitalization. While there is no impact on the upper-middle and high-income groups; NRCMS has substantially improved the capacity of low-income rural families to prevent poverty due to illness, especially for the lower-middle-income group. |
| Ranson et al. 2007 [79] *Published* | Cluster RCT | Household survey (2005) | 37,442 | Regression | NR | Relative to the uninsured members in their subdistricts of residence, the mean socioeconomic status of SEWA-enrolled households increased significantly, on average by 6.9 on the 100 points scale (p<0.001). |
| Rao et al. 2009 [80] *Published* | Cluster RCT | Household survey (2004) | 320 | Bivariate analysis | CHF members used health services more frequently than non-members. In all cases, the share of monthly visits made by members is disproportionately high compared with enrolment. | NR |

(*Continued*)

**Table 2.** (Continued)

| Study Publication status | Study design | Data sources (Data year) | Sample size | Identification strategy | Key findings on the impact on healthcare utilization | Key findings on the impact on financial protection |
|---|---|---|---|---|---|---|
| Ravit et al. 2020 [81] *Published* | Cross-sectional | Multiple Indicator Cluster Survey (MICS) (2015) | 4,172 | PSM and Regression | CBHI-enrolment increases the probability of having at least one antenatal care (ANC) by 13% (95% CI: 10–15%; P<0.001) and the probability of having four or more ANCs by 11% (95% CI: 6–16%; P<0.001). CBHI-enrolled women are 15% more likely (95% CI: 10–19%; P<0.001) to give birth at a healthcare facility), and delivery with qualified staff by 8% (95% CI: 4–12%; P<0.001). However, CBHI enrolment had no significant impact on Caesarean section rate or postnatal care. | NR |
| Raza et al. 2016 [82] *Published* | Stepped-wedge cluster RCT | Household panel data (2010–2013) | 3,685 | Regression | CBHI enrolment did not show a significant effect on utilization of both outpatient and inpatient care in the three study settings | Conditional on use, CBHI enrolment reduced outpatient expenses but increase inpatient expenses, there is no evidence that CBHI enrolment reduced OOP expenditures or on the probability of hardship financing in all three districts. |
| Robyn et al. 2012 (A) [83] *Published* | Stepped-wedge cluster RCT | Household panel data (2004–2006) | 705 | Regression | CBHI enrolment did not significantly increase access to seek treatment in general. **AOR = 1.331 (0.518– 2.144)** | NR |
| Robyn et al. 2012 (B) [84] *Published* | Cross-sectional | Household survey (2007) | 990 | PSM and Regression | CBHI-enrolment maintained a significant positive effect on the utilization of facility-based care versus uninsured. **ARR = 2.73 (SE = 1.26)** | NR |
| Sheth 2021 [85] *Published* | Cluster RCT | Household survey (2012) | 1,686 | Difference-in-differences | CBHI-enrolments does not lead to statistically significant increase in healthcare utilization compared with uninsured households. | CBHI-enrolments does not have a statistically significant effect on financial protection (assessed by selling of assets or levels of debt, eroding savings, and borrowing). |

(*Continued*)

**Table 2.** (Continued)

| Study Publication status | Study design | Data sources (Data year) | Sample size | Identification strategy | Key findings on the impact on healthcare utilization | Key findings on the impact on financial protection |
|---|---|---|---|---|---|---|
| Shimeles 2010 [86] *Working paper* | Cross-sectional | Household survey (2006) | 6,900 | Instrumental variable | CBHI membership significantly increased health care utilization by about 15% following an illness episode. Higher utilization of health care services was found among the insured non-poor (about 26.9%) than insured poor households (about 8.5%) | CBHI membership significantly reduced household's catastrophic health expenditure (10% THE) by about 16.4% accessing medical care. Slightly higher protection was found among the insured non-poor (about 24%) than insured poor households (about 23%) |
| Simieneh et al. 2021 [87] *Published* | Case-comparison | Household survey (2016) | 410 | PSM and Regression | CBHI membership contributed a 28.70% increase to healthcare utilization compared with non-members ($p < 0.001$) **AOR = 3.12 (1.94–5.02)** | NR |
| Sun et al. 2009 [88] *Published* | Before-after controlled | Household survey (2005) | 3,101 | Matching | NR | Financial protection from the China's New Cooperative Medical Scheme was modest **40% NFE** **AOR = 0.909 (0.764–1.083)** |
| Tilahun et al. 2018 [89] *Published* | Cross-sectional | Household survey (2016) | 652 | PSM and Regression | Being a member to mutual health insurance contributed approximately 25.2% point increase healthcare utilization. **AOR = 2.16 (1.45–3.23)** | NR |
| Wagstaff 2007 [96] *Working paper* | Cross-sectional | Vietnam Household Living Standard Survey (2004) | 9,000 | PSM, Difference-in-differences, and Regression | Vietnam's HCFP increased the probability of an inpatient spell by 30%, but the probability of an outpatient visits by only 16%. However, this improved utilization was least impactful on the poorest decile. | The HCFP increased out-of-pocket (OOP) payments for both outpatient visits and inpatient care. Although, the scheme reduced the risk of catastrophic OOP spending by 3–4 percentage points, 32% of beneficiaries still experienced catastrophic OOP spending. |
| Wagstaff et al. 2009 [97] *Published* | Cross-sectional | Household panel data (2003–2005) | 8,476 (28,696 persons) | PSM and Regression | NCMS has had an appreciable positive impact on the utilization of both outpatient (ATT = 0.060) and inpatient services (ATT = 0.020). | NCMS significantly increased the cost of outpatient visits (ATT = 10.80) and inpatient admissions (ATT = 5.00) but reduced the cost of deliveries (ATT = -12.33). Less evident impact on poorest quintile. |

*(Continued)*

**Table 2.** (Continued)

| Study Publication status | Study design | Data sources (Data year) | Sample size | Identification strategy | Key findings on the impact on healthcare utilization | Key findings on the impact on financial protection |
|---|---|---|---|---|---|---|
| Woldemichael & Shimeles 2015 [90] *Working paper* | Cross-sectional | Rwandan Integrated Household Living Condition Survey (2010/2011) | 14,308 (63,398 persons) | Bayesian adjustments (Rubin Causal Model) | CBHI-membership significantly increased the likelihood of utilizing medical consultation services and screening services but not utilization of drugs. Impact was highest among married women & under-five children | NR |
| Woldemichael et al. 2019 [91] *Working paper* | Cross-sectional | RwandanIntegrated Household Living Conditions Survey (2000, 2005, and 2010) | 6,390 (2000) 6,259 (2005) 13,546 * (2010) | Bayesian adjustments (Rubin Causal Model) | NR | CBHI-enrolment increases the probability but decreases the amount of out-of-pocket (OOP) spending for overall health services. The average treatment effect is higher for richer quintiles. However, for outpatient services, it increases both probability and amount of OOP spending |
| Yang 2015 [92] *Published* | Pre-post study | China Health and Nutrition Survey (2009) | 1,846 | Bivariate analysis | NR | There was no difference in catastrophic health payment and health payment-induced impoverishment after NCMS reimbursements **5% THE AOR = 0.920 (0.680–1.245) 10% THE AOR = 0.915 (0.669–1.251) 15% THE AOR = 0.922 (0.668–1.273) 20% THE AOR = 0.932 (0.670–1.296) 25% THE AOR = 0.942 (0.672–1.321)** The poorest quintile disproportionately experienced catastrophic health spending. |
| Yilma et al. 2015 [93] *Published* | Cross-sectional | Household panel survey (2011 to 2013) | 1,632 | PSM and Regression | NR | CBHI enrolment reduced reliance on potentially harmful coping responses such as borrowing due to lost income during ill health CBHI enrolment leads to a 5-percentage point—or 13%—decline in the probability of borrowing and is associated with an increase in household income during illness episodes. |
| Yip & Hsiao 2010 [94] *Published* | Pre-post treatment-controlled | Household panel survey (2004–2006) | 3,081 | PSM and Difference-in-differences | RMHC increased the probability of an outpatient visit by 0.12 ($p < 0.01$) and reduced the probability of self-medication | At the 15% NFE threshold, the RMHC reduced the rates of catastrophic health spending by 0.076 (from 0.245; $p < 0.05$) and reduced medical impoverishment by 0.129 (from baseline rate of 0.626, $p<0.05$) for those in the lowest quartile. |

*(Continued)*

**Table 2.** (Continued)

| Study Publication status | Study design | Data sources (Data year) | Sample size | Identification strategy | Key findings on the impact on healthcare utilization | Key findings on the impact on financial protection |
|---|---|---|---|---|---|---|
| Yip et al. 2008 [95] *Working paper* | Before-After Intervention Controlled | Household panel survey (2004–2006) | 2,726 | PSM and Difference-in-differences | RMHC has increased the probability of outpatient visits by 70% and reduced the probability of self-medication by similar percentages. There was no statistically significant impact on hospitalization. The lowest and highest-income individuals experienced the greatest increase in outpatient utilization. | NR |

**CBHI**: Community-based Health Insurance, **NFE**: Non-food expenditure, **NR**: Not reported, **THE**: Total household expenditure,

\* Sample size not included in the total, as it is already counted Woldemichael & Shimeles 2015

having a serious or critical overall risk of bias (S3 Fig). Of note, most included non-RCTs and quasi-experimental studies were rated as having a low risk of bias for the classification of participants, deviation from intended interventions, measurement of outcomes, bias in the measurement of outcomes, and selection of reported results. Based on weighted risk using study sample size (in households), 80% of these studies were rated as having a low risk of bias, 16% as having a moderate risk of bias, and about 4% as serious/critical risk of bias (S4 Fig). The main causes of serious overall bias risk, according to ROBINS-I assessment for non-RCTs and

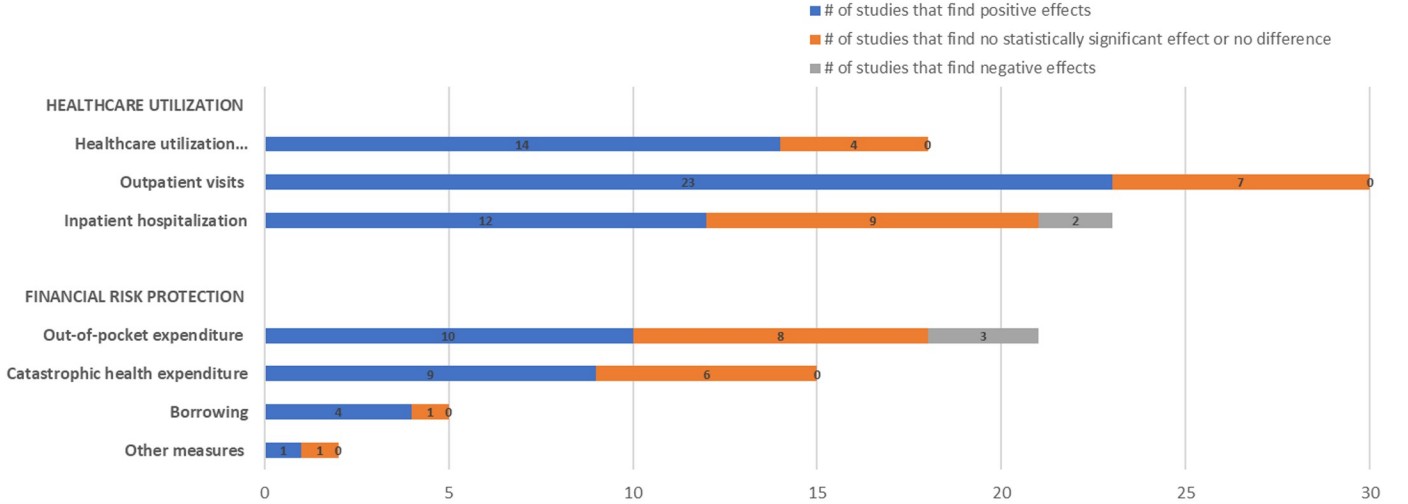

**Fig 3. Studies reporting the impact of CBHI schemes on healthcare utilization and financial risk protection in LMICs.**

quasi-experimental studies, were weaknesses in the confounding bias and selection of participants domains. Our assessment suggests these bias nudges the CBHI effects towards the null.

## Healthcare utilization

**Fig 3** summarizes the evidence of the impact of CBHI on the utilization of healthcare services. The evidence on utilization of healthcare services and outpatient services generally suggested a positive effect, with 14 out of 18 studies and 23 out of 30 studies reporting a statistically significant positive effect, respectively. However, the evidence on inpatient hospitalization is less clear, with 12 out of 23 studies reporting a positive effect, two studies finding a negative effect, and nine studies reporting statistically insignificant effects. Among the higher quality studies, that is, those with low overall risk of bias from RoB 2.0 and ROBINS-I assessments: 10 out of 13 studies, 20 out of 25 studies, and 10 out of 20 studies reported a positive relationship between CBHI enrolment and healthcare utilization, use of outpatient health services, and inpatient hospitalization, respectively. Two high-quality studies, however, reported CBHI enrolment decreased inpatient hospitalization [41, 77]. The existing evidence suggests that the government-supported community-involved model had the greatest impact on healthcare utilization–**Table 3**. However, the two studies showing a negative impact of CBHI on healthcare utilization were also a government-supported community-involved model in SSA, suggesting that government support in this region could have both favourable and unfavourable effects [41, 77]. In addition, the impact of CBHI on healthcare utilization generally improves as the scheme with older schemes–**Table 3**.

Meta-analysis of data from included studies showed that CBHI significantly improved overall healthcare utilization: AOR = 1.64 (95% CI = 1.12–2.39, $I^2$ = 79.1%, n = 4 studies, sample = 5,122 households); use of outpatient medical services: AOR = 1.58 (95% CI = 1.22–2.05, $I^2$ = 89.2%, n = 7 studies, sample = 42,210 households), and health facility delivery (maternity): AOR = 2.21 (95% CI = 1.61–3.02, $I^2$ = 53.6%, n = 2 studies, sample = 7,140 households)–**Fig 4**, respectively. However, pooled data suggests CBHI had insignificant improved inpatient hospitalization: OR = 1.53 (95% CI = 0.74–3.14, $I^2$ = 81.3%, sample = 2,886 households)–**Fig 4**. A CBHI scheme that's exclusive for pregnant mothers decreased the Caesarean section delivery rate (AOR = 0.42, 95% CI = 0.22–0.78) [77]. Of note, restricting these analyses to higher quality studies yielded similar results. Likewise, restricting these analyses to non-China studies also yielded similar results.

Sensitivity analysis leaving individual studies, including all studies from China, did not yield significantly different impact results. In subgroup analysis, we did not find statistically significant differences in CBHI impact size for healthcare utilization based on a country's income status (p = 0.61), region (p = 0.23), CBHI model (p = 0.20), and study quality (p = 0.20). However, the impact sizes were significantly different considering the study design. RCTs reported a slightly lower pooled estimate (OR = 1.12, 95% CI = 1.05–1.21) than non-RCTs and quasi-experiments (OR = 2.13, 95% CI = 1.62–2.80), p < 0.001 –**S3 Table**. For the utilization of outpatient medical services, however, there were no significant differences by income study CBHI model (p = 0.93) nor study quality (p = 0.95), but the impact estimates were statistically different by publication status (p = 0.01), region (p < 0.01), and stay design (p < 0.01). In this context, RCTs reported a substantially higher impact estimate (OR = 3.99, 95% CI = 2.53–6.27) than non-RCTs and quasi-experiments (OR = 1.55, 95% CI = 1.24–1.94)–**S4 Table**.

**Table 3. CBHI impact on healthcare utilization and financial risk protection by CBHI mode, duration of scheme, and region.**

| | Healthcare utilization (N = 71) | | | | Financial risk protection (N = 43) | | | |
|---|---|---|---|---|---|---|---|---|
| | Positive impact | No impact | Negative impact | Total | Positive impact | No impact | Negative impact | Total |
| **Region** | | | | | | | | |
| ∘ East Asia and Pacific | 13 (72.2%) | 5 (27.8%) | 0 (0.0%) | 18 | 6 (40.0%) | 6 (40.0%) | 3 (20.0%) | 15 |
| ∘ Southeast Asia | 5 (50.0%) | 5 (50.0%) | 0 (0.0%) | 10 | 5 (83.3%) | 1 (16.7%) | 0 (0.0%) | 6 |
| ∘ Sub-Saharan Africa | 31 (72.1%) | 10 (23.3%) | 2 (4.6%) | 43 | 13 (59.1%) | 9 (40.9%) | 0 (0.0%) | 22 |
| **China vs non-China studies** | | | | | | | | |
| ∘ Studies from China | 8 (57.1%) | 6 (42.9%) | 0 (0.0%) | 14 | 3 (27.3%) | 5 (45.4%) | 3 (27.3%) | 11 |
| ∘ Studies from outside China | 41 (71.9%) | 14 (24.6%) | 2 (3.5%) | 57 | 21 (65.6%) | 11 (34.4%) | 0 (0.0%) | 32 |
| **CBHI model** | | | | | | | | |
| ∘ Provider-based model | 4 (80.0%) | 1 (20.0%) | 0 (0.0%) | 5 | 1 (100.0%) | 0 (0.0%) | 0 (0.0%) | 1 |
| ∘ Community-driven and community-managed model | 16 (59.3%) | 11 (40.7%) | 0 (0.0%) | 27 | 10 (55.6%) | 8 (44.4%) | 0 (0.0%) | 18 |
| ∘ Government-supported community-involved model | 29 (74.4%) | 8 (20.5%) | 2 (5.1%) | 39 | 13 (54.2%) | 8 (33.3%) | 3 (12.5%) | 24 |
| **Duration of CBHI scheme** | | | | | | | | |
| ∘ < 3 years | 4 (44.4%) | 5 (55.6%) | 0 (0.0%) | 9 | 3 (50.0%) | 3 (50.0%) | 0 (0.0%) | 6 |
| ∘ 3–10 years | 32 (71.1%) | 11 (24.5%) | 2 (4.4%) | 45 | 15 (51.7%) | 11 (37.9%) | 3 (10.4%) | 29 |
| ∘ > 10 years | 13 (76.5%) | 4 (23.5%) | 0 (0.0%) | 17 | 6 (75.0%) | 2 (25.0%) | 0 (0.0%) | 8 |
| **Household members covered** | | | | | | | | |
| ∘ Women and children | 13 (72.2%) | 4 (22.2%) | 1 (5.6%) | 18 | 1 (100.0%) | 0 (0.0%) | 0 (0.0%) | 1 |
| ∘ Entire household | 36 (67.9%) | 16 (30.2%) | 1 (1.9%) | 53 | 23 (54.8%) | 16 (38.1%) | 3 (7.1%) | 42 |
| **Study setting** | | | | | | | | |
| ∘ Rural | 26 (70.3%) | 11 (29.7%) | 0 (0.0%) | 37 | 14 (50.0%) | 11 (39.3%) | 3 (10.7%) | 28 |
| ∘ Urban and semi-urban | 2 (66.7%) | 1 (33.3%) | 0 (0.0%) | 3 | 1 (100.0%) | 0 (0.0%) | 0 (0.0%) | 1 |
| ∘ Both rural and urban | 21 (67.7%) | 8 (25.8%) | 2 (6.5%) | 31 | 9 (64.3%) | 5 (35.7%) | 0 (0.0%) | 14 |
| **Study design** | | | | | | | | |
| ∘ Randomized controlled | 3 (33.3%) | 6 (66.7%) | 0 (0.0%) | 9 | 4 (57.1%) | 3 (42.9%) | 0 (0.0%) | 7 |
| ∘ Non-randomized | 46 (74.2%) | 14 (22.6%) | 2 (3.2%) | 62 | 20 (55.6%) | 13 (36.1%) | 3 (8.3%) | 36 |

CBHI: Community-based health insurance

## Financial risk protection

Overall, the evidence on the impact of CBHI on financial risk protection is less consistent than that for healthcare utilization–**Fig 3**. In total, 21 of the 61 studies reported the impact of CBHI

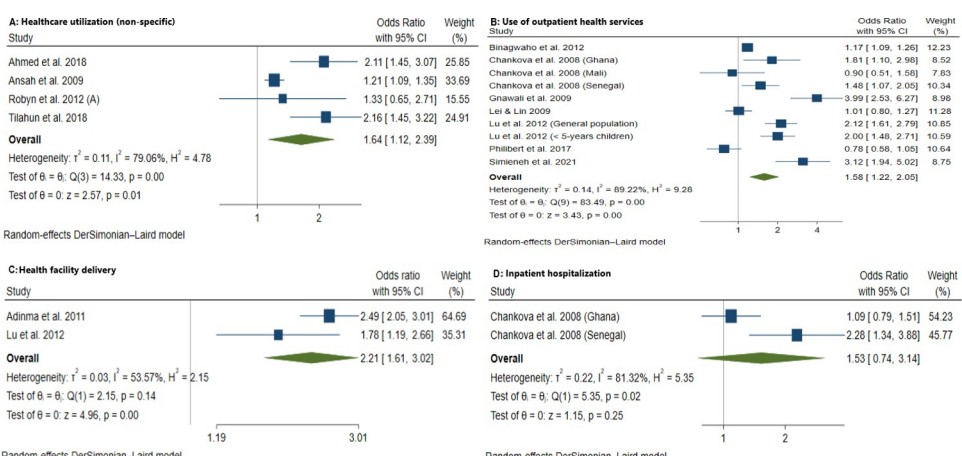

**Fig 4.** Pooled estimate for impact of CBHI on (A) Healthcare utilization (non-specific), (B) Use of outpatient health services, (C) Health facility delivery, and (D) Inpatient hospitalization. CI: Confidence interval.

on the level of OOP health expenditure. Among those 21 studies, 10 found a positive effect (that is, a reduction in OOP expenditure), eight studies found no statistically significant effect, and three studies–all from China [52, 53, 97]–reported a negative effect (that is, an increase in OOP expenditure). Another financial protection measure is the probability of incurring catastrophic health expenditure, defined as OOP payments exceeding a certain threshold percentage of total expenditure, income, non-food expenditure, or capacity-to-pay. Of the 14 studies reporting this measure, nine reported reductions in the risk of catastrophic expenditure and six found no statistically significant difference. Four of the five studies that used borrowing as a measure of financial protection reported a positive impact of CBHI [46, 56, 65, 93], whereas a single study reported no impact [54]. Finally, two high-quality studies evaluated the impact on financial protection by assessing the impact of CBHI on household assets and the

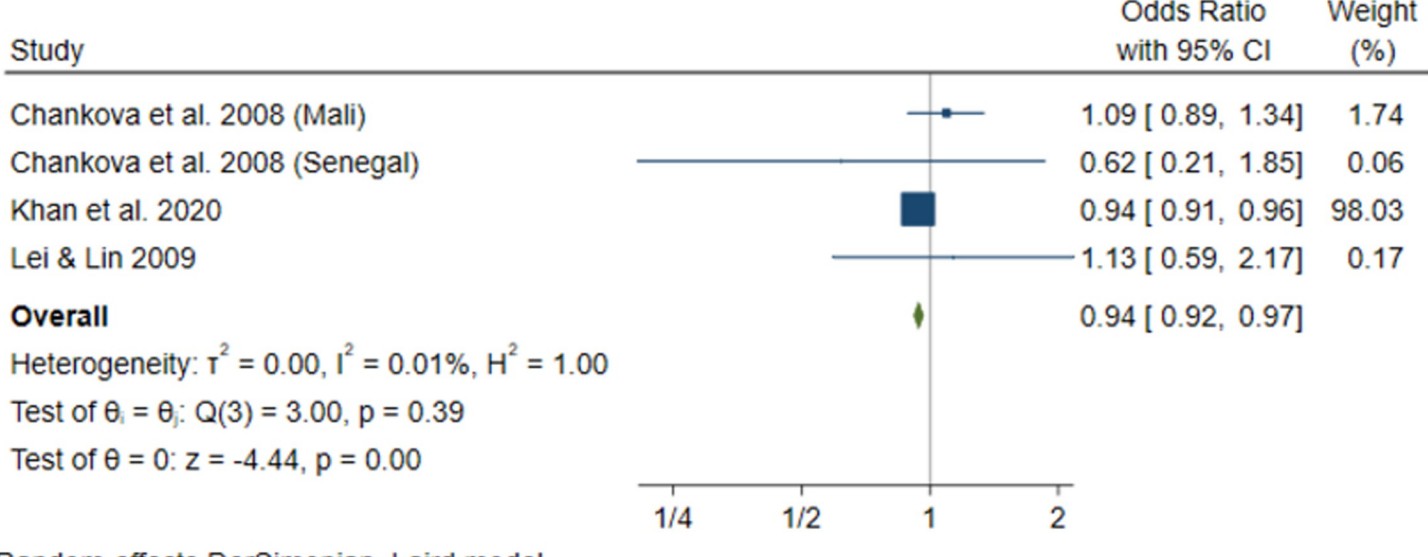

**Fig 5. Pooled estimate for impact of CBHI on OOP expenditure.** CI: Confidence interval.

probability of falling into poverty [75, 78]. Although CBHI had no significant impact on household assets [75], it significantly decreased the probability of falling into poverty [78].

Among high-quality studies, that is, those with low overall risk of bias from RoB 2.0 and ROBINS-I assessment, 8 out of 16 studies, 8 out of 13 studies, and 4 out of 4 studies reported a positive relationship between CBHI enrolment and decrease in the level in OOP health expenditure, decrease in the incidence of catastrophic expenditure, and decrease in the probability of borrowing. In addition, the impact of CBHI on financial risk protection generally improves over time–**Table 3**. The lone study that evaluated the impact of CBHI on financial risk protection exclusively in urban and semi-urban setting showed the positive impact of CBHI [42], but only half of studies conducted in rural settings showed a positive outcome. Likewise, the lone study that evaluated the impact of CBHI on financial protection for only women and children showed that CBHI provides significant financial risk protection [98]. Studies employing both RCT and non-RCT study designs largely reported similar results on the impact of CBHI on financial risk protection.

Meta-analyses of data from included studies showed that CBHI significantly decreased the level of OOP health expenditure: AOR = 0.94 (95% CI = 0.92–0.97, $I^2$ = 0.0%, n = 4 studies, sample = 8,983 households); reduced the incidence of catastrophic health expenditure at 10% total household expenditure threshold: AOR = 0.69 (95% CI = 0.54–0.88, $I^2$ = 59.6%, n = 4 studies, sample = 10,614 households) and 40% non-food expenditure threshold: AOR = 0.72 (95% CI = 0.54–0.96, $I^2$ = 76.6%, n = 4 studies, sample = 22,543 households)–**Figs 5 and 6**. Restricting these meta-analyses to higher quality studies yielded similar results. Likewise, restricting these analyses to non-China studies also yielded similar results. Leave one sensitivity study also yielded largely analogous results. In subgroup analysis, there were no significant differences in CBHI impact size on the level of OOP expenditure by national income status (p = 0.29), region (p = 0.38), CBHI model (p = 0.63), and study's quality (p = 0.21)–**S5 Table**. There was also no difference in the incidence of catastrophic health incidence at 10% total household expenditure threshold by national income (p = 0.38), region (p = 0.61), CBHI model (p = 0.31), and study design (p = 0.31); and at 40% non-food expenditure threshold by national income (p = 0.24), region (p = 0.09), publication status (p = 0.52), and study quality (p = 0.23)–**S6 and S7 Tables**.

## Quality of evidence

The GRADE assessments for the impact of CBHI schemes on overall healthcare utilization and financial risk protection in LMICs are outlined in **S8 Table**. The certainty of evidence varied. The evidence for overall healthcare utilization, use of outpatient services, and health facility delivery were graded as high, but the evidence for inpatient hospitalization was assessed as low. However, the evidence for the impact of CBHI on OOP health expenditure and catastrophic health expenditure were graded as high.

## Discussion

This systematic review has summarized the best available evidence from 61 unique studies on the impact of CBHI schemes on healthcare utilization and financial risk protection in LMICs. The evidence suggests that, compared to uninsured households, CBHI-insured households had improved utilization of healthcare services but only marginally improved financial protection while accessing care. The findings from the meta-analyses support these findings, even though fewer reports were included in the meta-analyses with large heterogeneity in these outcomes. The body of evidence studying the effectiveness of CBHI schemes, determinants of enrolment in these schemes, and factors associated with renewal of subscription has increased,

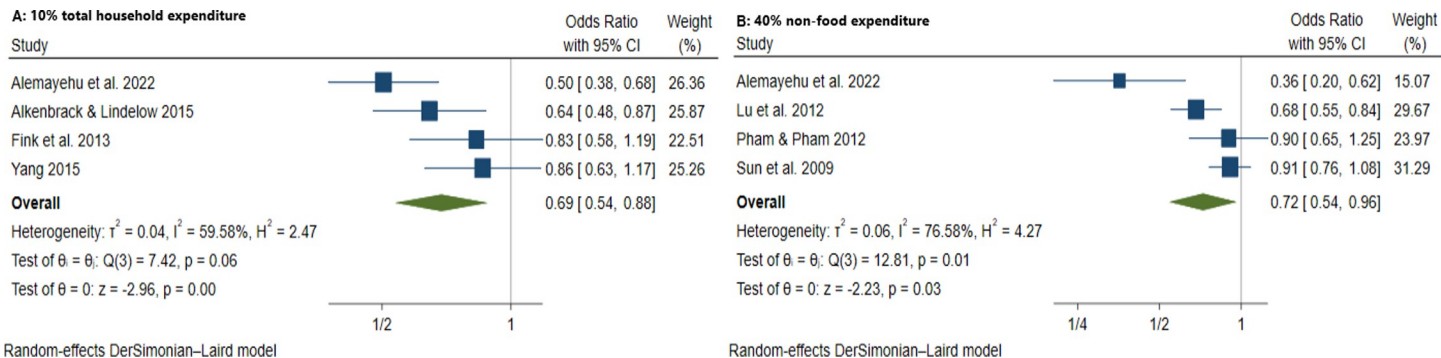

**Fig 6.** Pooled estimate for impact of CBHI on catastrophic health expenditure at (A) 10% total household expenditure, (B) 40% Non-food expenditure. CI: Confidence interval.

suggesting that health system researchers and policymakers find this topic relevant. This review has summarized and pooled available evidence and delivered strong conclusions on the CBHI impact for achieving financial risk protection and access to quality essential healthcare–key targets for UHC, while signalling an effective operational model that warrants further research.

Previous works have specifically focused on the impact of CBHI on financial risk protection [16, 17], healthcare utilization [18, 19], or both [4, 20]. However, none of these reviews included a quantitative meta-analysis of the data nor addressed the selection bias inherent in evaluating health insurance during study selection. Although Bhageerathy et al. narratively conclude that CBHI enrolment increased access to healthcare facilities and improved utilization of healthcare services [19]. Artignan and Bellanger's review of CBHI schemes in SSA suggests this improvement was only evident for outpatient care but weakly evident for inpatient care [18]. Habib et al. review of the impact of micro health insurance on financial protection and narratively suggest a positive influence of MHI on OOP, catastrophic health expenditure, poverty, health expenditures, household consumption, borrowings, sale of assets, and household savings [16]. Previously, Ekman's narrative review showed the same conclusion [17]. Spaan et al.'s and Mebratie et al.'s reviews reached the same conclusion: the impact of CBHI on improving healthcare utilization–especially fairly cheaper outpatient care services–and mitigating catastrophic healthcare spending [4, 20].

Healthcare utilization is a key performance indicator for measuring universal health coverage and our study indicates that CBHI immensely improved healthcare utilization especially outpatient healthcare services. By reducing the monetary cost of accessing healthcare services, CBHI schemes may induce higher utilization–a term known as moral hazard [186]. However, this increased utilization represents overcoming barriers to accessing healthcare rather than wasteful healthcare consumption per se [187]. The extent to which CBHI schemes overcome this financial barrier could depend on the benefits package, coverage, and co-payment policies [51, 188]. CBHI schemes, like the one in Burkina Faso, offer a comprehensive benefit package with minimum exclusions and no co-payments remove uncertainties at the time of illness and are likely to increase utilization [58]. Our review, however, provides weak evidence for inpatient hospitalization. First, CBHI schemes that demonstrated no/negative impact of CBHI on inpatient hospitalization did not cover inpatient admission [51]. In addition, even when inpatient hospitalization is covered, hospital admission is decided by physicians whose medical assessments moderate the impact of CBHI enrolment. Furthermore, CBHI enrolment should, in the long term, reduce the need for inpatient hospitalization, as chronic illnesses requiring

hospital admission, are addressed early in outpatient clinics given the improved access to care [189].

CBHI is not only associated with higher healthcare utilization and better financial risk protection for enrolled households. The evidence regarding the protective effect of CBHI in LMICs, while not as strong as the evidence for healthcare utilization, is still positive. Increased volume and intensity of healthcare produces a smaller reduction in OOP expenditures than what it would have been otherwise [190], and in some instances erases the protective effects of insurance [52, 97]. This dovetails with the hypothesis that in LMIC, CBHI enrolment overcomes the financial barrier to access but fails to adequately protect the enrolled once inside the healthcare system [172]. The uneven success achieved in terms of providing financial risk protection is due to country-specific variations in the CBHI scheme implementation, benefit package, scheme coverage, and cost-sharing policy. However, delicately balancing affordable premiums, improving enrolment and coverage, providing generous benefits, and still remaining sustainable is often elusive for schemes without external funding and policy support. Schemes with fairly cheap premiums leave enrollees with high OOP expenses when they access care, especially for high-cost treatments [52, 188]. On the other hand, increasing premiums to reduce cost-sharing makes enrollment impossible for poor households–the main target of CBHI schemes. Nevertheless, our study findings establishes that CBHI provides significant financial risk protection for enrolled households.

## Study limitations

To the best of our knowledge, our review is the most comprehensive analysis to date of the causal impact of CBHI on healthcare utilization and financial risk protection in LMICs. We also addressed the selection bias and heterogeneity issues–a common weakness in previous reviews, by including primary studies that addressed selection bias through randomization or appropriate statistical techniques. Hence, our conclusions are based on primary studies with causal inferences. Furthermore, we performed meta-analyses to provide more robust evidence that can help researchers and policymakers better understand the magnitude of the impact. Our study has several limitations. First, all systematic reviews are susceptible to publication and selection bias. Ours is not different, even though we minimized these biases by employing a comprehensive, pre-registered search strategy developed with the assistance of a university librarian, searching through multiple databases and grey literature, and utilizing two independent reviewers for study identification. Second, we had an unequal representation of countries that affects our findings' generalizability–a quarter of included studies (14 out of 61 studies) were from China. The absence of eligible studies from LAC, MENA, and Europe and Central Asia regions exacerbates this limitation. However, sensitivity analyses excluding these studies did not yield different results. Third, as data from only a few studies were included in meta-analyses, we employed descriptive content analysis, which involves greater reliance on the original authors' interpretations. Fourth, due to the limited number of studies included in the meta-analysis, we did not perform funnel plot tests to examine for heterogeneity, non-reporting bias, and chance in our pooled impact estimates [33]. However, to limit the chance that results from additional studies would be missing from our synthesis, we scrubbed through multiple specialized databases, searched grey literature, and considered studies in multiple languages.

## Policy implications

Consistent with a growing body of literature, our review provides strong evidence of the causal impact of CBHI on healthcare utilization and financial risk protection in LMICs. Our study

also provides compelling evidence that government-supported CBHI models improve health-care utilization and financial protection. This is crucial given that most households in rural communities and in the informal sector, which are the key targets of CBHI schemes, cannot afford premiums that can sustain the schemes. If the schemes are to offer comprehensive benefit package with minimum exclusions and co-payments, the need for external funding and policy support is even greater [8, 188]. Equally important, nesting CBHI schemes within pre-existing social institutions (such as market women association, tricycle riders association, etc.) is necessary (but insufficient) for successful implementation. Enduring schemes such as the Self-Employed Women Association scheme in India and *mutuelles de santé* in Senegal typifies this [51, 60, 79], as these schemes provide strong social cohesion and managerial expertise required to achieve insurance objectives.

## Conclusion

This systematic review and meta-analysis found evidence that CBHI improves healthcare utilization for enrolled households in LMICs. Although the evidence for financial risk protection is not as consistent as that for healthcare utilization, the evidence is still positive regardless of the health cost-induced catastrophe or impoverishment metric. This evidence, congruent with evidence from previous empirical studies, suggests that with a few pragmatic policy reforms and operational modifications, LMIC struggling to achieve UHC through publicly-funded health insurance schemes may consider CBHI for this purpose.

## Supporting information

**S1 Fig. Assessment plot of Cochrane RoB 2.0 risk of bias assessment and internal validity of included randomized controlled trials.**
(TIF)

**S2 Fig. Summary plot of Cochrane RoB 2.0 risk of bias assessment and internal validity of included randomized controlled trials.**
(TIF)

**S3 Fig. Assessment plot of Cochrane ROBINS-I risk of bias assessment and internal validity of included non-randomized controlled trials.**
(TIF)

**S4 Fig. Summary plot of Cochrane ROBINS-I risk of bias assessment and internal validity of included non-randomized controlled trials.**
(TIF)

**S1 Text. Search strategy.**
(DOCX)

**S1 Table. Eligibility criteria. Inclusion criteria for studies reporting impact of community-based health insurance (CBHI) schemes on healthcare utilization and financial risk protection in low- and middle-income countries (LMICs).**
(DOCX)

**S2 Table. Retrieved full text articles/studies excluded from the review and reasons for exclusion.**
(DOCX)

**S3 Table. Sub-group analysis of the impact of CBHI on healthcare utilization (non-specific) in LMICs.**
(DOCX)

**S4 Table. Sub-group analysis of the impact of CBHI on outpatient health services in LMICs.**
(DOCX)

**S5 Table. Sub-group analysis of the impact of CBHI on OOP health expenditure in LMICs.**
(DOCX)

**S6 Table. Sub-group analysis of the impact of CBHI on catastrophic health expenditure at 10% total household expenditure threshold in LMICs.**
(DOCX)

**S7 Table. Sub-group analysis of the impact of CBHI on catastrophic health expenditure at 40% non-food expenditure threshold in LMICs.**
(DOCX)

**S8 Table. GRADE assessment for quality of evidence.**
(PDF)

**S1 Checklist. PRISMA 2020 checklist.**
(DOCX)

## Author Contributions

**Conceptualization:** Paul Eze, Lucky Osaheni Lawani.

**Data curation:** Paul Eze, Stanley Ilechukwu, Lucky Osaheni Lawani.

**Formal analysis:** Paul Eze, Stanley Ilechukwu, Lucky Osaheni Lawani.

**Investigation:** Paul Eze, Lucky Osaheni Lawani.

**Methodology:** Paul Eze, Stanley Ilechukwu, Lucky Osaheni Lawani.

**Project administration:** Paul Eze.

**Software:** Paul Eze.

**Supervision:** Paul Eze.

**Validation:** Paul Eze, Lucky Osaheni Lawani.

**Visualization:** Paul Eze.

**Writing – original draft:** Paul Eze.

**Writing – review & editing:** Paul Eze, Stanley Ilechukwu, Lucky Osaheni Lawani.

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
