## [Decision Letter · Decision Letter 0]

4 Apr 2023

PONE-D-22-32762Impact of community-based health insurance in low- and middle-income countries: A systematic review and meta-analysisPLOS ONE

Dear Dr. Eze,

Thank you for submitting your manuscript to PLOS ONE. After careful consideration, we feel that it has merit but does not fully meet PLOS ONE’s publication criteria as it currently stands. Therefore, we invite you to submit a revised version of the manuscript that addresses the points raised during the review process.

We look forward to receiving your revised manuscript.

Kind regards,

Lara Vojnov

Academic Editor

PLOS ONE

Journal Requirements:

2. We note that Figure 2 in your submission contain [map/satellite] images which may be copyrighted. All PLOS content is published under the Creative Commons Attribution License (CC BY 4.0), which means that the manuscript, images, and Supporting Information files will be freely available online, and any third party is permitted to access, download, copy, distribute, and use these materials in any way, even commercially, with proper attribution. For these reasons, we cannot publish previously copyrighted maps or satellite images created using proprietary data, such as Google software (Google Maps, Street View, and Earth). For more information, see our copyright guidelines: http://journals.plos.org/plosone/s/licenses-and-copyright.

NASA Earth Observatory (public domain): http://eart hobservatory.nasa.gov/   

Additional Editor Comments :

Dear Authors,

Thank you for submitting your manuscript to PLoS One. We have reviewed and while not yet acceptable for publication hope that you will carefully reflect the reviewer comments (below) and resubmit once addressed.

Reviewers' comments:

Reviewer's Responses to Questions

**Comments to the Author**

1. Is the manuscript technically sound, and do the data support the conclusions?

Reviewer #1: Yes

Reviewer #2: Yes

2. Has the statistical analysis been performed appropriately and rigorously? 

Reviewer #1: Yes

Reviewer #2: Yes

3. Have the authors made all data underlying the findings in their manuscript fully available?

Reviewer #1: Yes

Reviewer #2: Yes

4. Is the manuscript presented in an intelligible fashion and written in standard English?

Reviewer #1: Yes

Reviewer #2: Yes

5. Review Comments to the Author

Reviewer #1: This is a very useful review and contribution to the literature. It can be further improved by a few additional elements in the analysis

1. The authors report the effects on health care utilization but do not detail any of the information on data sources; how much was gleaned from surveys versus other forms of health care utilization? were all trials equal? any disaggregation by age groups impacted (effects on women, children and elderly)?

2. Is there further evidence of impact by gender or geography? I note the higher impact in public sector or government led programs but were any differences notable by urban or rural settings?

3. One notes the effects were not significant on inpatient utilization? does that reflect maternity care as well?

4. Noting the bulk of studies being from China, did the authors conduct a sensitivity analysis with and without these data?

Reviewer #2: This is a nice paper. I am not an expert in meta-analysis so the paper should be reviewed by a statistician. However, overall the paper is extremely well written and detailed with close attention to detail

• Table 2 is mislabeled as Table 3

• I suggest being explicit about financial risk protection measures under the “data analysis” paragraph rather than waiting to results section to describe

• Metanalysis Figures seem very low resolution and are hard to read. This may be an issue with the submission platform compressing files for review but will need to be doublechecked

• Figure 4-6 readability would be better if labels “A” “B” etc were replaced by descriptive labels.

• I think the phrase "even though several indicated otherwse” could be eliminated from the abstract

• I’d also suggest eliminated the "The governmentsupported community-involved model relatively had the greatest impact” reference in the abstract; since this taxonomy isn’t introduced here it is hard to understand

• The last two paragraphs of the conclusion (discusing health utilization and financial risk) could be a bit more definitive about what this study shows. Study findings are interwoven with literature in a way that doesn’t immediately make clear how the study clears up some of the debates in the literature. The paragraph on financial risk in particular is mostly negative on the impact of CBHI despite the postiive findings of the meta-analysis reported here.

6. PLOS authors have the option to publish the peer review history of their article (what does this mean?). If published, this will include your full peer review and any attached files.

Reviewer #1: **Yes: **Zulfiqar Ahmed Bhutta

Reviewer #2: **Yes: **Peter Rohloff

---

## [Author Response · Author response to Decision Letter 0]

10 May 2023

Manuscript Number: PONE-D-22-32762

Title: Impact of community-based health insurance in low- and middle-income countries: A systematic review and meta-analysis

Dear Dr. Lara Vojnov, PhD

Academic Editor

PLOS ONE

We are very grateful to the editorial team and the reviewers for their useful comments on our manuscript. Their suggestions have greatly enhanced the quality of our manuscript. All modifications and corrections have been made as recommended. Below are our responses to the reviewers’ comments.

Journal requirements 

#1. Please ensure that your manuscript meets PLOS ONE's style requirements, including those for file naming. The PLOS ONE style templates can be found at https://journals.plos.org/plosone/s/file?id=wjVg/PLOSOne_formatting_sample_main_body.pdf and https://journals.plos.org/plosone/s/file?id=ba62/PLOSOne_formatting_sample_title_authors_affiliations.pdf.

Authors’ response 

We have made all required changes to meet PLOS ONE’s style requirements. 

#2. We note that Figure 2 in your submission contain [map/satellite] images which may be copyrighted. All PLOS content is published under the Creative Commons Attribution License (CC BY 4.0), which means that the manuscript, images, and Supporting Information files will be freely available online, and any third party is permitted to access, download, copy, distribute, and use these materials in any way, even commercially, with proper attribution. For these reasons, we cannot publish previously copyrighted maps or satellite images created using proprietary data, such as Google software (Google Maps, Street View, and Earth). For more information, see our copyright guidelines: http://journals.plos.org/plosone/s/licenses-and-copyright

The following resources for replacing copyrighted map figures may be helpful: USGS National Map Viewer (public domain): http://viewer.nationalmap.gov/viewer/; The Gateway to Astronaut Photography of Earth (public domain): http://eol.jsc.nasa.gov/sseop/clickmap/; Maps at the CIA (public domain): https://www.cia.gov/library/publications/the-world-factbook/index.html and https://www.cia.gov/library/publications/cia-maps-publications/index.html; NASA Earth Observatory (public domain): http://earthobservatory.nasa.gov/; Landsat: http://landsat.visibleearth.nasa.gov/; USGS EROS (Earth Resources Observatory and Science (EROS) Center) (public domain): http://eros.usgs.gov/#; Natural Earth (public domain): http://www.naturalearthdata.com/

Authors’ response 

Thanks very much for this information. We actually obtained our base layer map from Natural Earth and included a statement credit to Natural Earth in the manuscript (see page 20). 

Reviewer #1

#1. The authors report the effects on health care utilization but do not detail any of the information on data sources; how much was gleaned from surveys versus other forms of health care utilization? were all trials equal? any disaggregation by age groups impacted (effects on women, children and elderly)?

Authors’ response 

Thanks for this recommendation. We have provided more information on study design and age groups in Table 3 and in the results section.

#2. Is there further evidence of impact by gender or geography? I note the higher impact in public sector or government led programs but were any differences notable by urban or rural settings.

Authors’ response 

Thanks very much for this recommendation. We have provided evidence of impact by setting (rural or urban) in Table 3 and in the results section, but we could not extract evidence of impact in included studies by gender. 

#3. One notes the effects were not significant on inpatient utilization? does that reflect maternity care as well?

Authors’ response 

No, we noted that CBHI improved maternity inpatient utilization. However, the only study that specifically looked at Caesarean delivery showed that CBHI decreased Caesarean delivery rate – highlighted in blue (page 6).

#4. Noting the bulk of studies being from China, did the authors conduct a sensitivity analysis with and without these data?

Authors’ response 

Thanks for this observation. Yes, we conducted sensitivity analysis with and without these data from China and did not obtain different results for both healthcare utilization and financial risk protection outcomes. We have noted these in the results and discussion sections 

Reviewer #2

#1. Table 2 is mislabeled as Table 3

Authors’ response 

Thank you for this observation. We have corrected this. 

#2. I suggest being explicit about financial risk protection measures under the “data analysis” paragraph rather than waiting to results section to describe

Authors’ response 

Thanks very much for this suggestion. We have provided more details on the financial risk protection measures under the “data analysis” paragraph. 

#3. Metanalysis Figures seem very low resolution and are hard to read. This may be an issue with the submission platform compressing files for review but will need to be doublechecked

Authors’ response 

We double checked our meta-analysis figures and they are actually clear; we think it is the submission platform compressing the files for review. 

#4. Figure 4-6 readability would be better if labels “A” “B” etc were replaced by descriptive labels.

Authors’ response 

Thanks for the recommendation. We have included descriptive labels to improve readability. 

#5. I think the phrase "even though several indicated otherwise” could be eliminated from the abstract

Authors’ response 

Thanks for the recommendation. Done. 

#6. I’d also suggest eliminated the "The government-supported community-involved model relatively had the greatest impact” reference in the abstract; since this taxonomy isn’t introduced here it is hard to understand

Authors’ response 

Thanks for the recommendation. Done. 

#7. The last two paragraphs of the conclusion (discussing health utilization and financial risk) could be a bit more definitive about what this study shows. Study findings are interwoven with literature in a way that doesn’t immediately make clear how the study clears up some of the debates in the literature. The paragraph on financial risk in particular is mostly negative on the impact of CBHI despite the positive findings of the meta-analysis reported here.

Authors’ response 

Thank you very much for this observation. We were careful to avoid exaggerating the impact of CBHI, but we have revised these paragraphs to be clearer on the positive findings. 

We are very thankful to the PLOS ONE editorial team and reviewers for the extremely valuable recommendations. 

Thanks very much.

Paul Eze MD., MPH

Tel: +1 223 216 1640 (USA) 

University email: peze@psu.edu / Personal email: peze247@yahoo.com

On behalf of all the authors: 

1. Dr. Paul Eze, Pennsylvania State University, University Park, USA 

2. Dr. Stanley Ilechukwu, London School of Hygiene and Tropical Medicine, London, UK., and South Sahara Development Organization (SSDO), Enugu, Nigeria. 

3. Professor Lucky Osaheni Lawani, University of Toronto, Toronto, Canada

---

## [Editor Report · Decision Letter 1]

8 Jun 2023

Impact of community-based health insurance in low- and middle-income countries: A systematic review and meta-analysis

PONE-D-22-32762R1

Dear Dr. Eze,

We’re pleased to inform you that your manuscript has been judged scientifically suitable for publication and will be formally accepted for publication once it meets all outstanding technical requirements.

Kind regards,

Lara Vojnov

Academic Editor

PLOS ONE
---

## [Editor Report · Acceptance letter]

15 Jun 2023

PONE-D-22-32762R1 

Impact of community-based health insurance in low- and middle-income countries: A systematic review and meta-analysis 

Dear Dr. Eze:

I'm pleased to inform you that your manuscript has been deemed suitable for publication in PLOS ONE. Congratulations! Your manuscript is now with our production department. 

Kind regards, 

on behalf of

Dr. Lara Vojnov 

Academic Editor

PLOS ONE